# Stable and dynamic representations of value in the prefrontal cortex

**Pierre Enel[1,2]\*, Joni D Wallis[3,4], Erin L Rich[1,2]**

[1]Nash Family Neuroscience Department, Icahn School of Medicine at Mount Sinai, New York, United States; [2]Friedman Brain Institute, Icahn School of Medicine at Mount Sinai, New York, United States; [3]Helen Wills Neuroscience Institute, University of California at Berkeley, Berkeley, United States; [4]Department of Psychology, University of California at Berkeley, Berkeley, United States

**Abstract** Optimal decision-making requires that stimulus-value associations are kept up to date by constantly comparing the expected value of a stimulus with its experienced outcome. To do this, value information must be held in mind when a stimulus and outcome are separated in time. However, little is known about the neural mechanisms of working memory (WM) for value. Contradicting theories have suggested WM requires either persistent or transient neuronal activity, with stable or dynamic representations, respectively. To test these hypotheses, we recorded neuronal activity in the orbitofrontal and anterior cingulate cortex of two monkeys performing a valuation task. We found that features of all hypotheses were simultaneously present in prefrontal activity, and no single hypothesis was exclusively supported. Instead, mixed dynamics supported robust, time invariant value representations while also encoding the information in a temporally specific manner. We suggest that this hybrid coding is a critical mechanism supporting flexible cognitive abilities.

**\*For correspondence:**
pierre.enel@mssm.edu

**Competing interests:** The authors declare that no competing interests exist.

## Introduction

Theories of value-based decision-making suggest that the brain computes values for different choice options in order to compare items with different qualities on a common scale (*Padoa-Schioppa, 2011*). This attribution of a value relies on the association between an option and its outcome, and is learned by comparing the expected value of an option to the actual outcome experienced with it. However, when the presentation of an option is separated in time from its outcome, expected values must be temporarily held in memory for such comparisons to be possible.

The prefrontal cortex (PFC) plays a key role in working memory (WM), the temporary maintenance of information across such temporal gaps. While the lateral PFC is particularly involved in WM relating to cognitive information (*Constantinidis and Klingberg, 2016*), it is less clear how value expectations are maintained across delays. The content model of WM postulates that each prefrontal region maintains and manipulates the type of information that it is specialized to process (*Goldman-Rakic, 1996*; *Lara et al., 2009*). From this view, maintaining an accurate estimation of expected values may rely more on those regions of PFC involved in learning and decision-making, such as the orbitofrontal cortex (OFC) or anterior cingulate cortex (ACC). OFC and ACC unit activity is known to reflect expected values associated with stimuli or actions (*Rushworth and Behrens, 2008*; *Kennerley et al., 2009*; *Amiez et al., 2006*), but little is known about the dynamics in either region that could bridge delays between outcomes and their predictive cues.

Here, we aimed to assess how value information is maintained across task delays in OFC and ACC. Neural mechanisms that maintain other domains of information in WM have been widely studied, but remain a subject of debate (*Constantinidis et al., 2018*; *Lundqvist et al., 2018*). One theory suggests that persistent and stable neuronal activity maintains WM representations across time

(*Wang, 2001*; *Riley and Constantinidis, 2015*; *Constantinidis et al., 2018*). This hypothesis stems from the observation of consistent activity patterns in the PFC of monkeys while they wait during a delay to execute an action (*Funahashi et al., 1989*; *Fuster, 1973*). However, in various WM tasks, little persistent activity is found in the lateral lPFC of macaque monkeys (*Stokes et al., 2013*; *Lundqvist et al., 2016*; *Rainer and Miller, 2002*), yet the memoranda can be decoded throughout the delay (*Barak et al., 2010*). This prompted the development of a second theory, suggesting that WM involves dynamic representations and activity-silent synaptic encoding (*Stokes et al., 2017*; *Miller et al., 2018*). Yet another perspective postulates that WM could be implemented by the sequential activation of single neurons passing on information to bridge a delay (*Rajan et al., 2016*; *Harvey et al., 2012*).

Given these multiple perspectives, we investigated the nature of value representation in the OFC and ACC in a value-based decision-making task in which monkeys were presented a reward predicting cue associated with a reward delivered after a delay. While delay activity encoded value, attempts at categorizing activity as persistent or transient, stable or dynamic, failed to exclusively support either view. Instead, both characteristics could be found in single unit and population activities, such that a purely stable or dynamic representation of value could be extracted. We hypothesize that these mixed dynamics occurring in the same neural population serve unique purposes. Stable representations allow downstream regions to read out memoranda with the same decoding scheme irrespective of the delay duration, while dynamic representations encode temporal information necessary to anticipate events and prepare behaviors. From this view, a rich mixed dynamical regime could supply underlying mechanisms that support flexible cognitive abilities.

## Results

### Behavior

Two monkeys (subjects M and N) performed a value-based decision-making task in which visual stimuli were associated with juice rewards (*Figure 1A*). Images predicted an amount and type of reward, which was either primary in the form of juice at the end of the trial, or secondary as a bar that increased proportionally to the value associated with the stimulus (*Figure 1B*). The bar, always present on the screen, represented an amount of juice that was received when it was cashed in every four trials. A total of eight pictures comprised the set of stimuli associated with reward, with four different values matched between primary (juice) and secondary (bar) rewards in terms of the monkeys' preferences (*Rich and Wallis, 2016*). After the presentation of the pictures, monkeys were required to move a joystick either left or right depending on another visual cue in order to obtain the predicted reward. Joystick direction was unrelated to the reward-predicting picture, but if the incorrect response was made no reward was delivered.

Monkeys were presented with choice trials (~20% of trials), in which the animals could choose between two stimuli presented simultaneously on the screen. ChoicesSelections were made by fixating the stimulus of their choice, and the associated reward was delivered at the end of the trial. Both monkeys learned to perform the task optimally by selecting the stimulus associated with higher values in more than 90% of the trials (*Rich and Wallis, 2016*). A logistic regression model predicting the cues chosen by the animals with variables value and type showed a much lower coefficient for type, and its p-values were many orders of magnitude larger, confirming that the type of reward had little influence on the choice of animals compared to value (*Figure 1C*, coefficients with [p-value of t-test] for monkeys M and N respectively; type left/right: .41 [2.53e-3]/-.51 [1.75e-4] and −0.61 [2.6e-3]/0.32 [0.11], value left/right: 1.77 [6.1e-117]/−1.95 [3.02e-136] and 2.66 [1.59e-94]/−2.46 [2.27e-90]).

To assess the dynamics of value representations, we focused on the single cue trials only. Here, reaction times in the unrelated joystick task decreased as the value of the stimulus increased, providing evidence that monkeys maintained an expectation of the upcoming reward across the intervening time interval (*Rich and Wallis, 2016*, *Figure 1B*). Reaction times were also unaffected by the type of reward anticipated (linear regression predicting reaction time as a function of value and type in single cue trials : coefficients for value [p-value of t-test] for monkeys M and N, respectively; type: −21 [.08] and 14.5 [.32], value: −63.9 [1.73e-31] and −65.7 [2.66e-23]). Therefore, we focus mainly

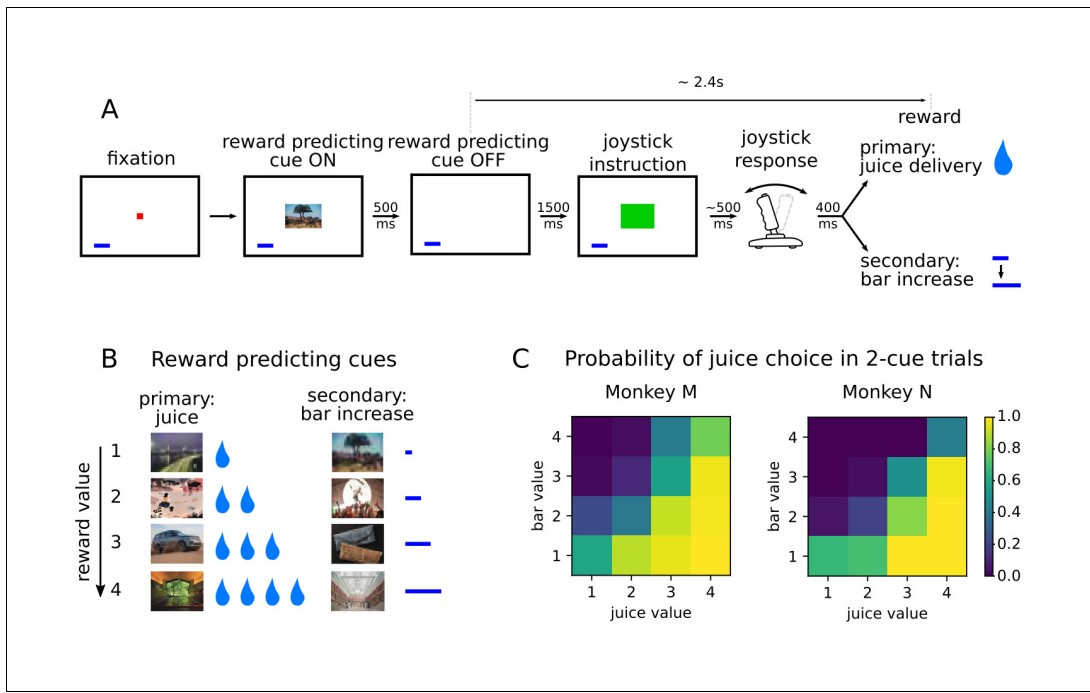

**Figure 1.** Value-based decision making task. (**A**) Monkeys initiated a trial by fixating a point in the center of the screen. A reward predicting cue appeared that the subject was required to fixate for 450 ms. After a 1500 ms delay, one of two possible images instructed the monkey to move a joystick right or left. Contingent on a correct joystick response, monkeys received a reward either in the form of juice (primary) or an increase of a reward bar (secondary), which was constantly displayed on screen (blue bar in figure). Note that the presentation of the reward predicting cue and the delivery of the reward are separated in time by more than 2 s. (**B**) A total of eight reward predicting pictures covered the combinations of four possible values, and two reward types. (**C**) Probability of choosing a juice option in choice trials for every pair of cues in which one predicts a juice reward and the other a bar reward.

on the representation of value during the period between the onset of the reward predicting cue and reward delivery, which lasted ~3 s, and included a ~2.4 s delay after the cue offset.

Single and multi units were recorded in the orbitofrontal (OFC) and anterior cingulate cortex (ACC) of the two subjects while they performed the decision-making task across 24 and 20 sessions (monkey M and N, respectively).

Units from all recording sessions and both monkeys were pooled together to create 2 populations of 798 and 315 neurons recorded from the OFC and ACC, respectively (*Table 1*). Basic unit and decoding results did not differ across monkeys (except for encoding and decoding of reward type in ACC, *Figure 2—figure supplement 1D and E*) so data were pooled.

## Encoding and decoding of value in unit activity

To assess the encoding of task variables by the neurons in these two regions, we regressed the firing rate of single and multi units on variables value, type and their interaction value x type and tested the contribution of these variables with an ANOVA. We considered value as a categorical variable, as some units activated in a non-linear fashion with respect to value, for example by responding only

**Table 1.** Number of recorded units with an average activity per session above 1 Hz.

| Monkey | Single M | Single N | Multi M | Multi N | Total |
|---|---|---|---|---|---|
| OFC | 259 | 192 | 170 | 177 | 798 |
| ACC | 114 | 72 | 81 | 48 | 315 |

to a specific value, and could not be modeled with a simple linear regression (*Figure 2—figure supplement 2*). Note that these activity profiles are not consistent with traditional and bprime;value coding', in which firing rates monotonically change with value, but they nonetheless encode value information. To demonstrate that a non-negligible proportion of units encoded value non-linearly, and that these units do not respond simply to the identity of one reward cue, we found the number of neurons significant for value in an ANOVA (activity ~ C(value) + type + C(value):type), that were not significant for an interaction between value and type with the same ANOVA, and not significant for value with a linear regression model (activity ~ value + type + value:type). Approximately 13% of units in both OFC and ACC fit these criteria.

Similar to previous reports (*Rich and Wallis, 2017*), the vast majority of OFC and ACC units encoded value at some point after the reward predicting cue and before the reward (73.2% for OFC and 77.8% for ACC, *Figure 2B*). The proportion of neurons encoding value peaked after the presentation of the reward predicting cue and steadily decreased until the reward (C, *Figure 2A*). The type of reward was only weakly encoded compared to value, with less than half the proportion of units encoding it at any time (29.8% for OFC and 37.1% for ACC, *Figure 2B*). No difference between

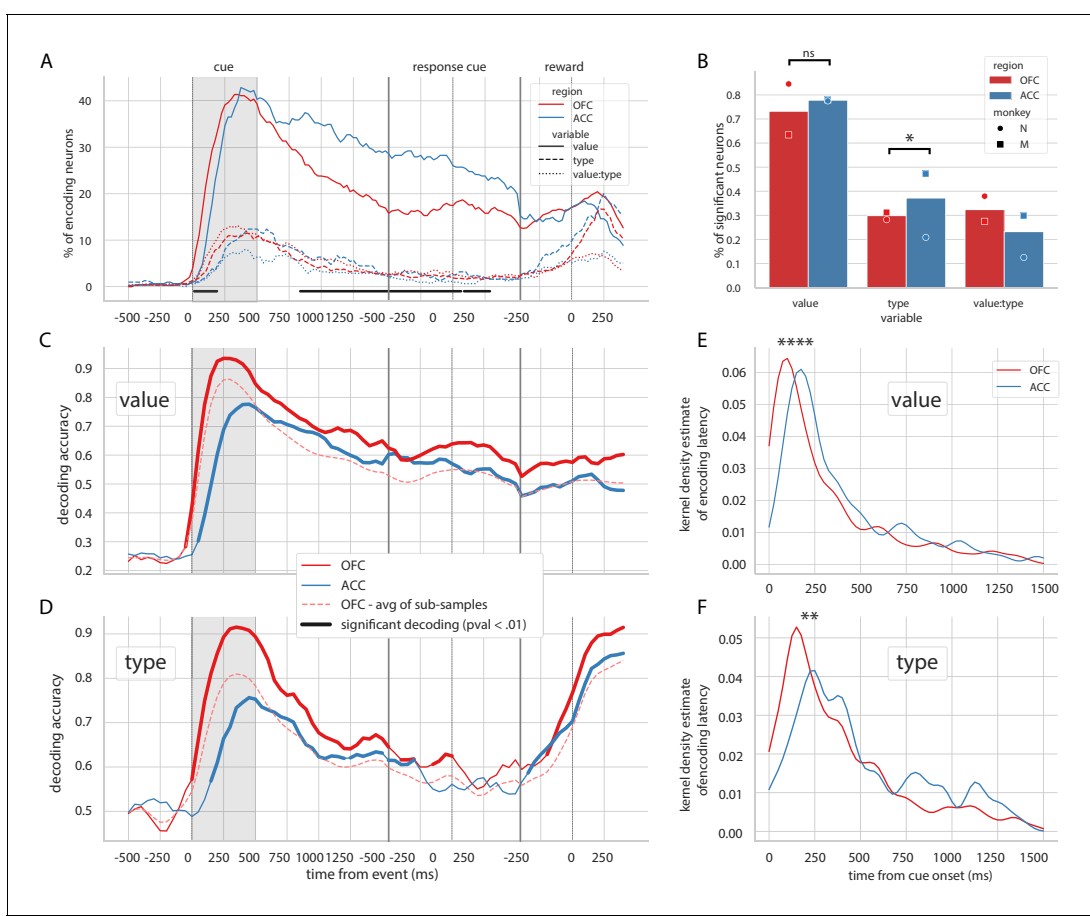

**Figure 2.** Encoding and decoding of task variables. (A) Percentage of units encoding the different variables at each point in time in both OFC and ACC. Significant difference in value encoding neurons with $\chi^2$ test is indicated by thick black lines at the bottom (p≤0.01). (B) Percentage of units encoding the task variables at any point in time from the onset of cue presentation to the delivery of the reward. (C and D) Average decoding accuracy of value and type, respectively, across five randomly generated population data sets, with a ridge regression classifier. Significant decoding is shown by the thick portions of line and corresponds to the time bins where the aggregated p-value of 1000-permutation tests for each of the five data sets was lower than 0.01. The dashed red line is the average of 200 sub-samples of OFC data with the same number of units as in ACC for comparison. (E and F) Kernel density estimate of encoding latencies in units for value and type, respectively. OFC latencies are significantly shorter (see main text).

The online version of this article includes the following figure supplement(s) for figure 2:

**Figure supplement 1.** Encoding and decoding of value and type for each monkey.
**Figure supplement 2.** Example neurons with nonlinear encoding of value.

ACC and OFC was found in the proportion of units encoding value at any point during the epoch of interest, but encoding of type was higher in ACC mostly driven by the data of one monkey (*Figure 2B*, $\chi^2$ test of independence, value: p=0.13 and type: p=0.022). The proportion of units encoding value was higher in ACC during the delay (*Figure 2A*); however, this difference was also mostly driven by the data from one monkey and is not significant in the other subject (*Figure 2—figure supplement 1A*). In accordance with previous findings (*Kennerley and Wallis, 2009*), the relative onset of value encoding was earlier in the OFC compared to ACC (*Figure 2E*). Latencies in the OFC were significantly shorter for both value and type (Kruskal-Wallis on latencies value:[stat = 19.4, p=1.05*10$^{-5}$] and type:[stat = 7.93, p=4.85*10$^{-3}$]). This result was replicated in each monkey's data independently (*Figure 2—figure supplement 1C*).

Population decoding of value with a linear ridge classifier elicited a peak in accuracy similar to the peak in the proportion of encoding units after the presentation of the delay, followed by a stable significant decoding until reward (*Figure 2C*, significance threshold set to p<0.01 with 1000 permutations, replicated in each monkey, *Figure 2—figure supplement 1B*). A similar result was observed for decoding type (*Figure 2C*), although decoding accuracy rapidly dropped and reached non significance around the joystick task. The first significant bin decoding value and type was 100 ms and 150 ms earlier respectively in OFC compared to ACC, and peak decoding was earlier and higher in OFC compared to ACC (for value/type: 93.5%/91.5% in OFC at 250 ms/350 ms, 77.6%/75.6% in ACC at 450 ms/450 ms).

Both regions had similar proportions of value encoding after the cue presentation; however, decoding accuracy was more than 10% higher in OFC, which can be partly explained by a higher unit count in the OFC population. To fairly compare decoding accuracy between the two regions, we randomly sub-sampled the OFC population 200 times to match the number of ACC units. The average accuracy of the sub-sampled populations was still slightly higher in OFC than ACC, indicating that the higher number of units in OFC alone cannot explain the difference in accuracy. The sensitivity index (also known as d') averaged across pairs of values in all neurons was higher in OFC (0.26) than in ACC (0.22), suggesting that values might be easier to separate in OFC activity compared to ACC, leading to higher decoding accuracy.

Together, these results show that value is strongly encoded in these prefrontal regions and can be significantly decoded throughout the delay. Because of the lower incidence of reward type encoding and lack of continuous type decoding throughout the delay, the rest of the study focused more on the representation of value than type.

## Persistent versus sequential encoding

The total number of neurons encoding value during the delay was much higher than the number encoding at any particular point in time, indicating that value was represented by different neurons throughout the delay. Given this, we next sought to understand the contribution of individual units to the population dynamics. A common method for doing this is sorting neuron responses. For example, to demonstrate that a neural population uses a sequential activation scheme, sorting the neurons by their peak activation typically displays a tiling of the delay (*Pastalkova et al., 2008*). To ensure that sequential encoding of value is a robust feature of OFC and ACC activity, we split the data into an equal number of training and testing trials. We sorted the units based on the time of peak value encoding during the delay with the training half, and found that both OFC and ACC value coding tiled the delay, such that different subsets of neurons were most selective at different times (*Figure 3A*). We then sorted the testing data according to the peak-encoding bin of the training data to assess its consistency (*Figure 3B*), and found that the tiling of the delay is also present in the testing data, although somewhat less clear. In the period restricted to the delay itself, peak encoding times did not follow a straight line, such that the proportion of encoding neurons decreased during the delay (*Figure 2A*). This suggests that encoding epochs are not uniformly distributed in time, consistent with a recent model of sequential encoding (*Rajan et al., 2016*). The peak encoding bins of training and testing data sets were also significantly correlated, which is expected with sequential encoding. Thus, our results partially support the hypothesis of a sequential representation of value in both areas, consistent with conclusions drawn from previous studies (*Pastalkova et al., 2008*), although the mere firing rate of units is often used (see *Figure 3—figure supplement 1A D* for firing rates). The same analyses of variable type yielded similar results (*Figure 3—figure supplement 2*).

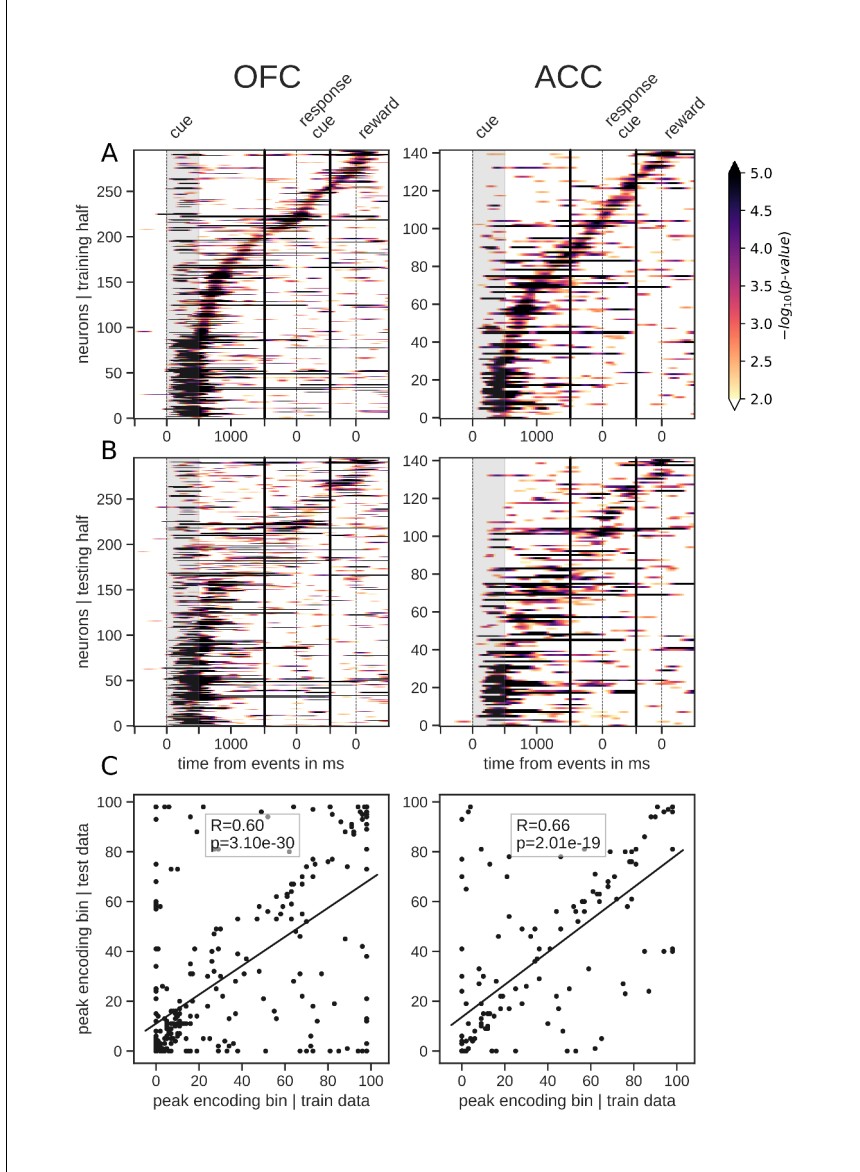

**Figure 3.** Tiling of the delay by value encoding units. (**A**) Tiling of individual units according to their peak encoding of value across the delay in both OFC and ACC with the training half of the trials. Colors represent the negative log of the encoding p-value (ANOVA, F test on value). The measure is bounded between 2 and 5 for visualization and corresponds to p-values ranging from 0.01 to $10^{-5}$. (**B**) Tiling with the testing half of the trials, with units sorted according to the training half of the trials, to show consistency in sequential encoding. Note that only units that had significant encoding in both the training and testing dataset were included, hence the lower number of units. (**C**) Scatter plot of the peak encoding bin for training versus testing half of the data. Graphs show the Spearman correlation coefficient, R, with associated p-value, p, and trendline.

The online version of this article includes the following figure supplement(s) for figure 3:

**Figure supplement 1.** Tiling and spanning of delay with unit firing rate.
**Figure supplement 2.** Tiling of the delay by reward type encoding units.

However, sorting the neurons by the proportion of the delay during which they encode value offers a different picture. *Figure 4* shows the same splitting procedure applied to encoding duration (*Figure 4—figure supplement 1* for variable type). While some units had time-limited encoding of value, others continuously encoded value throughout the delay, supporting the notion that there is a stable representation of value in the population. Similar results were obtained by clustering the

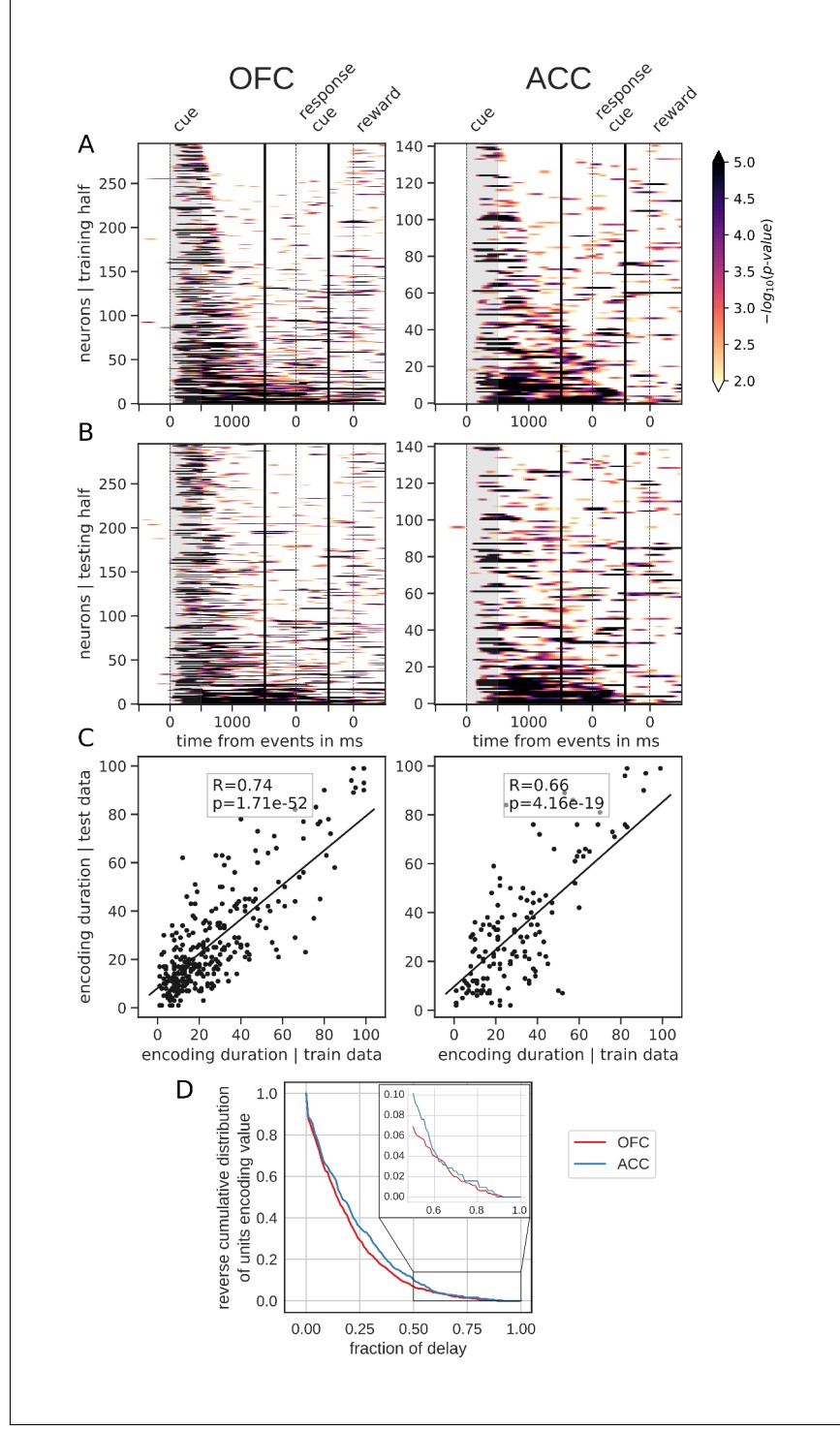

**Figure 4.** Spanning of the delay by value encoding units. (**A**) Units sorted by the duration of encoding of value with the training half of the trials. Colors represent the negative log of the encoding p-value (ANOVA, F test on value). The measure is bounded between 2 and 5 for visualization and corresponds to p-values ranging from 0.01 to $10^{-5}$. (**B**) Encoding durations of the testing half of the trials sorted by the duration the training trials. (**C**) Scatter plot of encoding duration for training versus testing half of the data. Graphs show the Spearman correlation coefficient, R, with associated p-value, p, and trendline. (**D**) Reverse cumulative distribution of units encoding value as a function of the fraction of the delay covered. This graph shows the proportion of units encoding value for at least the fraction of the delay indicated on the x-axis (no significant difference between regions, see main text). *Figure 4 continued on next page*

*Figure 4 continued*

The online version of this article includes the following figure supplement(s) for figure 4:

**Figure supplement 1.** Spanning of the delay by reward type encoding units.

average firing rates of neurons into two clusters (*Figure 3—figure supplement 1B E*). One cluster had increased firing rates for the duration of the delay, consistent with the persistent activity hypothesis. However, separating neurons into six clusters uncovered more diverse profiles of activity that nonetheless followed basic principles: phasic activity followed task events with stable or monotonically evolving activity between events (*Figure 3—figure supplement 1C F*). Thus, depending on the sorting method employed, the dynamic and sequential or stable and persistent hypothesis could be defended. Overall, encoding by most units covered a relatively small portion of the delay. A reverse cumulative distribution of the number of units encoding value showed a sharp decrease in the fraction of the delay covered (*Figure 4D*). Only 6.9% and 10.2% of units encoded value for more than half the delay in OFC and ACC, respectively, while less than 1% of neurons encoded type for that same duration (*Figure 4—figure supplement 1D*). Further, 80% coverage of the delay was found in roughly 1% of the units (OFC: 0.75%, ACC: 1.59%). The distribution of delay coverage by encoding units did not differ between OFC and ACC for the encoding of value, but there was a tendency for higher type encoding duration in ACC (Kolmogorov-Smirnov test, value: [stat = 0.079, p=0.90] and type:[stat = 0.21, p=0.025]).

## Population dynamics

Decoding analyses allow us to assess how well value information can be extracted from a population as a whole, but standard approaches that find unique solutions to optimally decode value at each point in time say little about the dynamics of a representation. In addition, our results show that value encoding among units can paint contrasting pictures at the level of population representation, that is, as both dynamic and sequential or stable and persistent. Therefore, to explore representational dynamics across time, we used cross-temporal decoding (CTD), in which a decoder is trained and tested at every possible pair of time bins in the delay (*Stokes et al., 2013*; *Astrand et al., 2014*; *Stoll et al., 2016*; *Meyers et al., 2008*). In this context, the classification accuracy is a proxy for the similarity in value representation between any two time points. We can then produce an accuracy matrix consisting of each pair of training/testing time bins. When the training and testing times are identical, we obtain a classical decoding procedure as shown in *Figure 2C*, which corresponds to the diagonal in the CTD accuracy matrix.

Using this method on the OFC and ACC populations, value representation dynamics were not clearly dynamic or stable (*Figure 5A*, *Figure 5—figure supplement 1A* for variable type). OFC evidenced a somewhat stable representation from 500 ms after the reward predicting cue (around cue offset) until the response cue, but the representation remained dynamic before and after this period, as the accuracy was mostly confined to the diagonal. ACC accuracy was lower overall but showed similar patterns. While significant decoding appeared to spread from reward cue to after the response cue, accuracy was not homogeneous and remained strongest along the diagonal, indicating a rather dynamic representation of value.

This dynamic pattern was also reflected in the population trajectory speed obtained by calculating the distance between two successive time bins in the neural space spanned by the estimated firing rate of each unit with a shorter smoothing window for more temporally accurate firing rate (50 ms SD Gaussian kernel, *Figure 6*). The reward predicting cue triggered an increase in speed that was followed by a period of low speed, indicative of relatively stable population activity, until the joystick instruction cue triggered another period of rapidly changing activity. Thus, strongly dynamic bouts of activity occur with changes in task stimuli, and in the absence of such changes, delay activity becomes quite stable.

## Stable representation of value

The basic CTD method relies on a decoder that weighs on the most discriminating features (i.e. neurons) at a particular point in time. If a sub-population strongly encodes value at one time, the decoder will attribute stronger weights to these units. However, if this sub-population changes

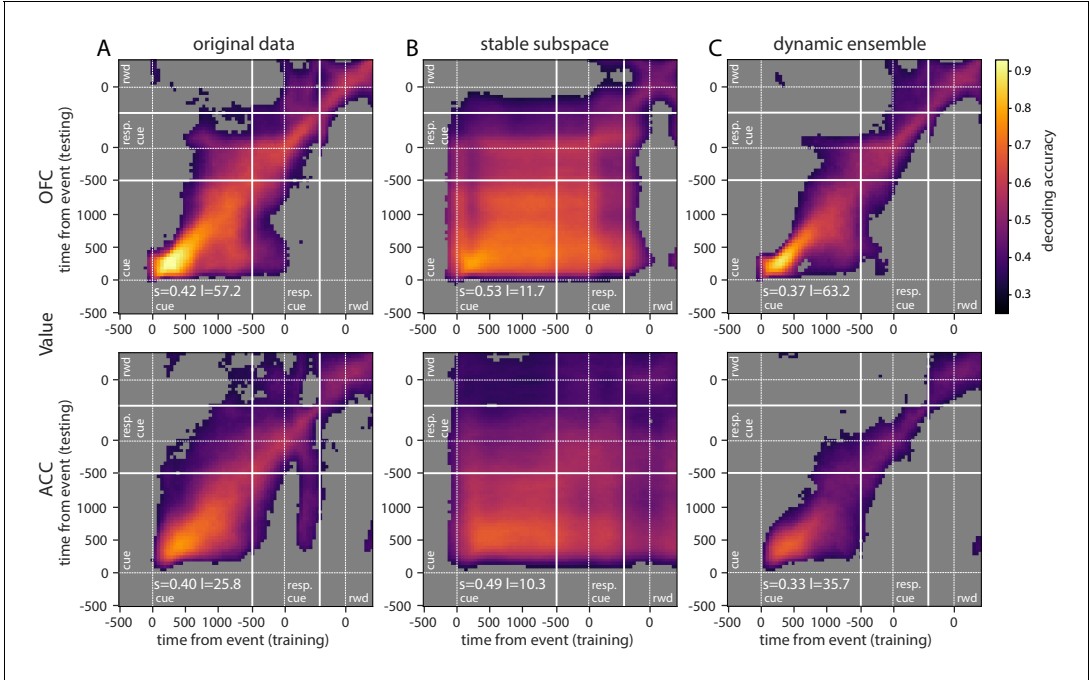

**Figure 5.** Accuracy of cross-temporal decoding (CTD) of value with different methods. Training and testing of a ridge classifier at different time bins across the delay. CTD was applied to (**A**) original data, (**B**) a stable subspace obtained from the combination of a value subspace and an ensemble method where units are iteratively removed to obtain maximum accuracy, and (**C**) on a dynamic ensemble where units were iteratively removed from the ensemble to maximize a dynamic score. Thick white lines indicate junctions between the three successive epochs (cue presentation, joystick task and reward delivery). The thin dashed white lines indicate the reference event of each epoch (reward predicting cue onset, response instruction, reward). The decoding accuracy corresponds to the average of five randomly generated pseudo populations and non-significant accuracy has been greyed out for clarity. The p-value threshold is 0.01 and p-values were obtained by aggregating the p-values of each dataset for a given training/testing time bin pair (see Methods section for more details; rwd = reward, resp. cue = response cue). The stable and locality scores are displayed on each panel (s = stability score, l = locality score). Note that significant decoding before the presentation of the cue is due to smoothing.

The online version of this article includes the following figure supplement(s) for figure 5:

**Figure supplement 1.** Cross-temporal decoding of reward type.

**Figure supplement 2.** Cross-temporal decoding of value with ensemble and subspace methods independently.

**Figure supplement 3.** Cross-temporal decoding of value for each monkey independently.

representations or ceases to encode value, the decoder accuracy will drop. Our results suggest such temporally local representations change across time, but this could simply be due to inhomogeneous representations of value at the population level. That is, the strongest representation at one time point may not generalize well to other time points. This would not exclude the possibility that a stable representation of value exists within the population, because the classifier is not optimized to find a decoding plane that is consistent throughout the delay.

To determine whether a linear combination of neural activities can be used to extract a stable representation of value, we combined two methods to optimize CTD. The first is a subspace method inspired by *Murray et al., 2017*, in which data are projected into a subspace that maximizes value representation while lowering the influence of temporal dynamics (see Materials and methods). This was applied to data from the onset of cue presentation to the delivery of reward. A three-dimensional subspace was obtained, and data projected onto this subspace was used to decode value, using CTD in the same manner as was done for the full population. Our cross-validation approach ensured that both train and test data were not used to generate the subspace, as it would introduce a bias in decoding results (see Materials and methods).

Repeating the CTD procedure with data projected onto the subspace demonstrates that more stable value representations can be extracted from the same population data (*Figure 5—figure supplement 2B*). Although significant decoding using CTD did not fully span the cue to reward period in OFC, higher accuracies were more widespread. ACC accuracy, which also spanned a larger

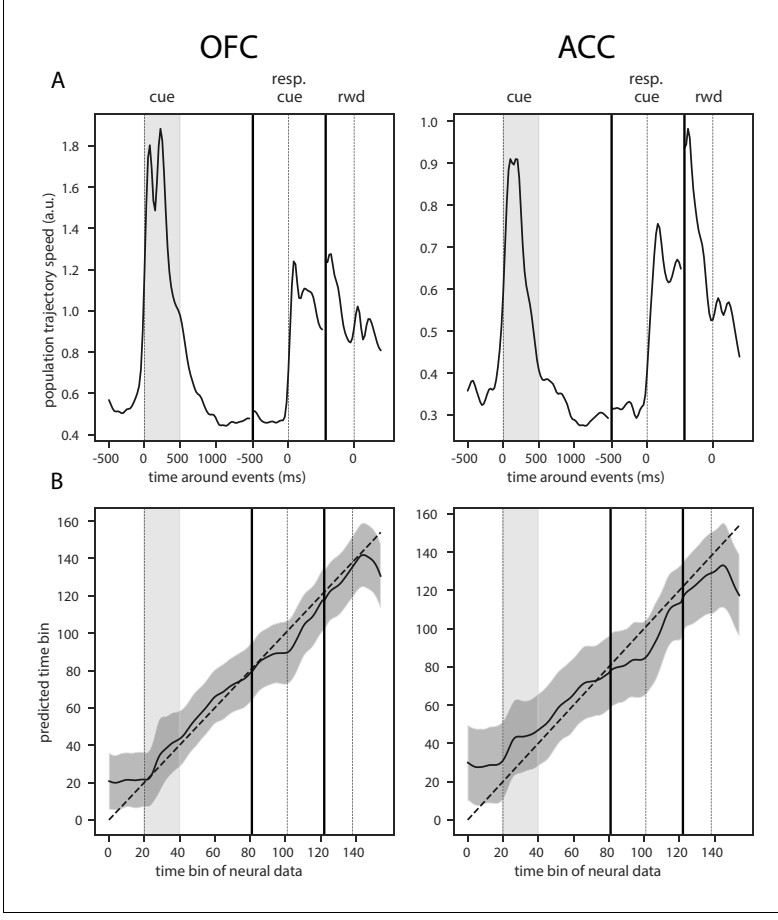

**Figure 6.** Trajectory speed and regression of time. (**A**) Speed (rate of change) of the average activity trajectory in the neural space defined by the full population of neurons, in OFC (left) and ACC (right). Note that the difference in scale between OFC and ACC (y-axis) is due to the difference in the number of units in each population. (**B**) Regression of the time bin from population activity with a simple linear regression. The black line and shaded area represent the average and standard deviation of leave-one-out cross-validation results. The dashed line corresponds to ground truth. For more temporally accurate results, the firing rate of each unit was estimated with a 50 ms standard deviation Gaussian kernel instead of the 100 ms used in the other analyses.

portion of the delay, covered nearly all of the epoch between cue and reward using the subspace approach.

Although a higher unit count is expected to produce higher decoding performances, in the case of CTD better performances might be obtained by pre-selecting the neurons that most participate in the stable decoding over time. For this purpose, we derived an ensemble method inspired by *Backen et al., 2018* that iteratively selects units based on their contribution to a stability score. This measure was defined as the averaged accuracy matrix of the CTD restricted to the delay period. Essentially, we started by removing each unit independently to find the n-1 ensemble that most increased the stability score, then removed a second unit to find the best n-2 ensemble and so on until only one unit was left, and then selected the ensemble that maximized decoding accuracy. This procedure markedly improved stable decoding across time, with significant decoding spanning all time pairs after the reward predicting cues (*Figure 5—figure supplement 2C*). The ensembles that best optimized the CTD accuracy in both regions contained only a fraction of the units from the full population (average of five randomly generated data sets: 107 out of 798 units for OFC and 71 out of 315 units for ACC). Greater numbers of units did not improve performance, likely because many of these modified their value encoding over time and did not provide a stable signal.

While the subspace and best ensemble approaches both found somewhat stable value representations, the most stability with highest accuracies was obtained with a combination of these

methods. The subspace procedure was applied to each ensemble selected by the iterative ensemble method. The result is a strikingly stable CTD in both regions for the entire duration of the delay, and beyond reward delivery in ACC (*Figure 5B*, *Figure 5—figure supplement 1B* for variable type). This demonstrates that when CTD is applied to a neural population, the representation may appear to be partially dynamic, yet it is possible to find a subspace of the population activity that defines a completely stable representation across time. In addition, combining subspace and ensemble procedures is a promising method to extract extremely stable representations of task variables. These results were replicated with data from each monkey independently (*Figure 5—figure supplement 3*).

## Dynamic representation of value

Since we found that stable value representations could be extracted with appropriate methods, it is logical that dynamic representations can also be identified. To do this, we first defined a 'locality' score to quantify how temporally local a representation is, based on the decoding accuracy over time. We fit a Gaussian curve to the decoding accuracy obtained from training at a given time bin and testing on all time bins. The height of the Gaussian divided by its standard deviation defined a score of temporal locality. The measure was defined as the average of the scores computed from the accuracy curves obtained across all training time bins. A dynamic representation of value was obtained by selecting the neurons with the ensemble method described above that maximized this measure. The resulting CTD accuracy displays the main feature of a strongly local and dynamic representation of value, where higher accuracy is confined to the diagonal (*Figure 5C*, *Figure 5—figure supplement 1C* for variable type). With the original full population, value representation in the OFC was rather stable in the middle of the delay, but the dynamic ensemble shrunk the higher accuracy block to constrain it to the diagonal. This result demonstrates that a dynamic subspace encoding local value representations can be extracted from the same neural population that encodes value in a very stable way. Together, these results support the idea that a basic CTD applied to a neural population yields a limited view of mixed neural dynamics.

Since the dynamic and stable ensembles yielded opposite dynamics, we expected limited overlap between them. Indeed, the overlap in units was found to be at chance levels for all ensemble instances obtained by cross-validation and data sets in both regions (25 instances from five-fold cross-validation and five data sets; $\chi^2$ test of independence with p≤0.01 and multi-test false discovery rate correction).

To further demonstrate that the activity in both of these areas is dynamic and supports the encoding of time, we regressed the index of time bins from 500 ms before the reward predicting cue onset until 500 ms after reward delivery with a simple linear regression from the neural population activity (*Figure 6B*). Indeed, time could be predicted accurately, supporting the notion of a robust representation of time in the neural population activity.

## Relationship between value encoding and decoding dynamics

To explore the relationship between single unit encoding and population dynamics, we correlated different encoding measures with their contribution to the value subspace and stable ensemble (Spearman correlation, see Materials and methods). We defined the encoding strength as the average of the negative log p-value of each unit across the delay, and encoding duration as the fraction of the delay when the unit significantly encoded value (p-value < 0.01). First we correlated each of these measures with the unsigned value subspace weights, since a stronger weight reflected a higher influence on the subspace irrespective of its sign (unsigned weights were averaged across five randomly generated population data sets). Both measures were highly correlated with subspace weights (*Figure 7A*, left and middle) indicating a clear link between encoding strength/duration and contributions to the value subspace. However, units might also change their pattern of value encoding during the delay. To quantify this, we defined an additional stability measure that includes the duration, strength and stability of value representation by penalizing cases where the difference in average firing between a pair of values changes sign (see Materials and methods for details). This measure was more strongly correlated with the value subspace weights (*Figure 7A*, right graph). Overall, these measures describe the qualities of activity that promote selection to the value subspace.

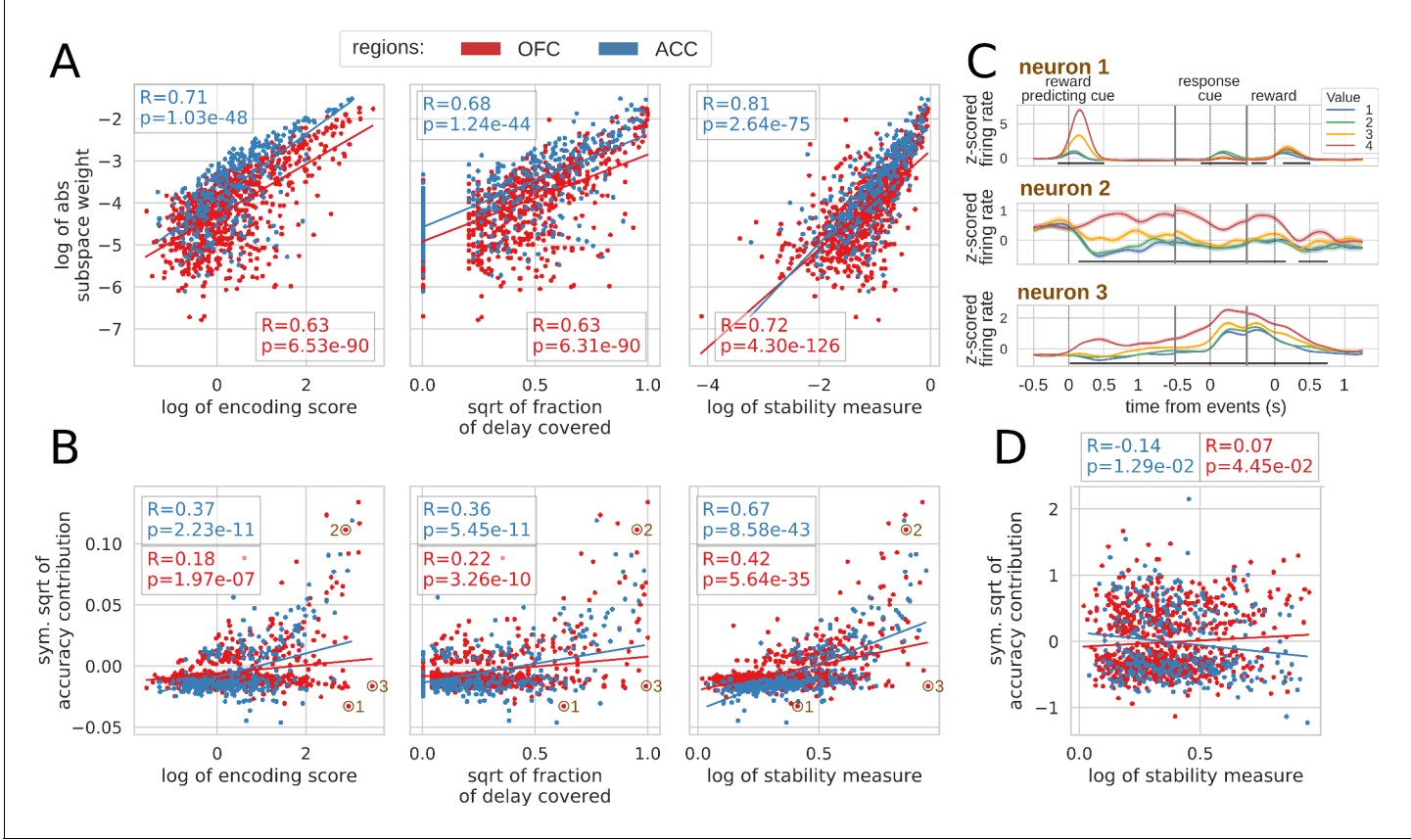

**Figure 7.** Correlation between encoding and decoding measures. (**A**) Correlations between encoding measures and subspace weights. (**B**) Correlations between encoding measures and CTD accuracy contribution. (**C**) Z-scored firing rate of two example neurons negatively contributing to the stable ensemble (1 and 3) and one neuron contributing positively (2). Individual data points corresponding to these neurons are circled in B. (**D**) Correlations between stability measure and locality measure contribution. All figures show Spearman correlations. sqrt = square root; negative contributions were transformed with a symmetrical function around the origin based on square root: $sign(x) * \sqrt{|x|}$. Transformations applied to the data had no effect on the correlations which are rank based (see Materials and methods).

To explore the contribution of each unit to the stable ensemble, we defined an accuracy contribution score based on the change in mean accuracy across all time bin pairs in the CTD elicited by removing a unit from the ensemble (this measure is an average across cross-validation and population data sets). This score was correlated with encoding strength, duration and stability score in both regions (*Figure 7B*, top row). A strongly encoding unit might encode value for a limited time and/or reverse its encoding (e.g. *Figure 7C* neuron 1), and a unit covering a large part of the delay might also change over time and not provide a stable representation. Conversely, units with stable encoding contribute more to the ensemble (*Figure 7C* neuron 2). However, since decoding relies on the interaction between many neurons, these simple encoding measures of individual units cannot fully explain all of the ensemble contributions. For example, neuron three in *Figure 7C* strongly encodes value for the whole delay, but contributes negatively to the stable ensemble.

Finally, contributions to the dynamic ensembles were not correlated with the stability measure (*Figure 7D*) as expected. However, these measures were neither anti-correlated which indicates that the contribution to a dynamic ensemble is not explained merely by an absence of stable encoding.

## Discussion

In this study, we shed light on the complex dynamics of value representations in prefrontal cortex. While value representations have been found in the activity of ACC and OFC (for example, *Rushworth and Behrens, 2008*), little is known about their dynamics across delays. One recent study found that OFC neurons that exhibit slow temporal integration also encode value more

strongly and do so from the time of a choice until the receipt of reward (*Cavanagh et al., 2016*). This suggested the presence of a subset of OFC neurons with stable encoding dynamics, consistent with the present results. Here, we expand on this to show that targeted methods elicit seemingly opposite results that do not lend themselves to straightforward interpretations with respect to the current theoretical frameworks common in the WM literature. Unit encoding shows features of both persistent and sequential activity, depending on which feature the analysis method is designed to extract. Similarly, targeted methods can extract either a stable subspace or a temporally local representation at the population level. These results, along with recent studies of the dynamic nature of delay activity in the prefrontal cortex (*Spaak et al., 2017*; *Murray et al., 2017*; *Meyers, 2018*), argue in favor of a more nuanced view than the pure persistent/stable, activity silent/dynamic or sequential activity hypotheses.

Here, we propose that mixed dynamical regimes, with both time and temporally insensitive information concurrently encoded in the same neural population, can provide a richer substrate to face varying task demands. On one hand, the main function of WM-related processes is to temporarily maintain information not available in the sensory environment so it can be manipulated or retrieved later. A stable neural representation is a robust means of holding such information online and is particularly useful when information must be extracted at an unspecified time. This is because a target population can reliably read out an unchanging representation whenever it is required. Conversely, representing relevant information in a time-sensitive manner is also critical, since most behaviors are organized in time, whether they are internally generated or aligned to external events (*Meyers, 2018*; *Cueva et al., 2019*). For example, our task includes a recurring sequence of events that the subjects had learned. By representing time, they can anticipate the next event and prepare to either process a task relevant stimulus or produce a motor output. Such preparatory processes can help optimize task performance and maximize the amount of reward the subject can receive.

These results present the coding of value in two dynamical regimes, either fully stable or fully dynamic. However, these are the representations that were specifically targeted by the present analysis methods, and it is possible that these dynamics are the extremes of a continuous spectrum of dynamical regimes that can be extracted by any downstream population, depending on the computational task it performs. Indeed, it has been shown that the time constant of neurons encoding value in prefrontal region is diverse (*Cavanagh et al., 2016*), supporting the hypothesis of a multitude of dynamical regimes. Similar to the idea that mixed selectivity provides a universal mix of task variables from which the representations most relevant to the task can be extracted (*Rigotti et al., 2013*), a multitude of dynamical regimes can provide a broad range of dynamics from which some can be selected for their relevance to the task (*Bernacchia et al., 2011*).

Beyond temporal dynamics, our data set also demonstrates notable heterogeneity in the representations of outcomes within these prefrontal networks. While value signals were robust in both OFC and ACC, consistent with previous reports of strong abstract value coding in these regions (*Padoa-Schioppa and Assad, 2006*; *Rushworth and Behrens, 2008*), there were also weaker representations of the anticipated reward type. This is consistent with findings in human and rodent OFC reporting value signals that are specific to a particular type of outcome (*Howard et al., 2015*; *Stalnaker et al., 2014*). Interestingly, this variable was barely encoded and decoded in ACC in one of the monkeys, suggesting that different information representation strategies were adopted by each subject. In addition, we provide evidence that neurons that encode value with linear changes in firing rate coexist with a smaller population of neurons that exhibit non-linear value encoding, some of which are non-monotonic, for example firing preferentially to value 3. Many internally generated continuous variables, such as time, space, and numerosity, are represented with both monotonic and non-monotonic activities in different brain regions (*Nieder and Dehaene, 2009*; *Eichenbaum, 2014*; *Wittmann, 2013*; *Moser and Moser, 2008*; *Funahashi, 2013*; *Dehaene and Brannon, 2011*) and our results demonstrate that value representations follow a similar format. Such nonlinear coding may enhance population-level signals (as it is the case with nonlinear mixed selectivity *Fusi et al., 2016*), and complements the rich representations of time and outcome type that are likely critical to accurately predicting multiple facets of an upcoming reward.

The debate about persistent versus dynamic representation in WM tasks has relied heavily on modeling experiments. Neural network models of persistent activity have historically been implemented as fixed point attractors when the remembered information comes from a discrete number of items (*Amit, 1995*; *Compte et al., 2000*) or bump/line attractors in the case of parametric WM

(*Machens et al., 2005*; *Seung, 1996*). Another neural network framework, with random recurrent connections referred to as reservoir computing, subsequently demonstrated that short term memory is readily obtained from generic networks in which different inputs (stimuli) elicit distinct trajectories thereby encoding stimuli for a short duration with highly dynamic signatures (*Maass et al., 2002*; *Jaeger, 2001*). While these dynamic properties appear irreconcilable, they are not necessarily exclusive and can be combined in a single network, as demonstrated by more recent models (*Maass et al., 2007*; *Pascanu and Jaeger, 2011*; *Enel et al., 2016*). Such networks can provide mixed dynamics similar to those observed in the present task, suggesting that they are better models for short-term value maintenance in OFC and ACC. Recent models also provide theoretical explanations for the degree of persistent versus dynamical activity. Specifically, some emerging theories suggest that this trade-off is a function of task requirements instead of the network architecture (*Masse et al., 2019*; *Orhan and Ma, 2019*). In addition, little is known about the effect of training on WM representations in experimental subjects such as monkeys. The degree of stable versus dynamic representations might vary depending on whether the subject is over-trained on the task or on particular stimuli in the task, such that over training might shift weak and dynamical activities toward more stable and persistent ones (*Barak et al., 2013*).

Among the dynamic representations of value, we found some evidence for sequential organization within our neural populations. Sequential activity is the successive, short-duration activation of individual neurons, a specific case of dynamical activity that follows a regular pattern. These patterns are thought to either pass information to bridge a delay or represent the passage of time in the absence of external stimuli (*Pastalkova et al., 2008*; *Rajan et al., 2016*). Sequential unit activity has also been associated with a variety of functions, including the temporal segmentation of memories in the hippocampus (*MacDonald et al., 2011*; *Eichenbaum, 2014*) and WM in parietal cortex and the rat medial PFC (*Harvey et al., 2012*; *Fujisawa et al., 2008*). To the best of our knowledge, this type of sequential encoding has not been reported in prefrontal regions of primates performing WM tasks, potentially because it has not been the focus of analyses to date. Consistent with the heterogeneous nature of our neural population, only some neurons showed evidence of reliable sequencing, while other neurons' activation and encoding was not short and uniform. Sequencing appeared most prominent early in the delay, and may coincide with the brief, highly dynamical epoch immediately after the presentation of a stimulus to be remembered that has been found in this and other studies. This dynamic period is typically followed by a more stable representation until the end of the delay (*Stokes et al., 2013*; *Murray et al., 2017*; *Barak et al., 2010*; *Wasmuht et al., 2018*; *Cavanagh et al., 2018*). It has been proposed that the dynamical epoch following the memorandum presentation is the result of stimulus processing, which in some tasks corresponds to the transformation of the stimulus into task-relevant information (*Stokes et al., 2013*), and our results suggest that this process may involve more reliable sequencing among neurons. More generally, it appears that task relevant inputs to a brain region trigger local processing that is reflected in transiently increased and dynamic activity as was demonstrated by a reservoir model mimicking ACC activity dynamics (*Enel et al., 2016*).

One challenge for attractor networks modeling WM tasks has been maintaining stable representations in the presence of distractors or intervening task demands. In the current study, monkeys had to perform a joystick sub-task before receiving reward, and CTD with the original neural populations yielded stable value representations until the presentation of the joystick instruction cue. Similar dynamics have been observed when a distractor perturbs the representation of a spatial cue held in WM (*Parthasarathy et al., 2017*; *Cavanagh et al., 2018*). The initially stable representation might be the result of unperturbed network activity that persists until a task relevant input (the response instruction cue) impacts the networks' dynamics. However, here we show that analyses designed to identify stable representations can, indeed, extract a subspace of the same population activity that was stable across the intervening joystick task. Interestingly, value encoding and basic decoding remained largely unaffected during the intervening task, unlike results previously reported in a dual task experiment (*Watanabe and Funahashi, 2014*). Here, when monkeys solved attention and memory tasks concurrently, selectivity for components of either task diminished at the level of single neurons and population activity. Similarly, intervening stimuli or distractors presented during the delay of a WM task disrupt the representation of memoranda (*Lebedev et al., 2004*; *Warden and Miller, 2007*; *Jacob and Nieder, 2014*). The discrepancy between these and our results might lie in the non-overlapping nature of value information and joystick task. In our case, the task and stimuli were

independent, whereas in these other studies, a stimulus of the same nature as the memoranda (e.g. a visual stimulus) was presented, perhaps triggering a conflicting representation.

A notable conclusion from this study is that OFC and ACC have similar dynamics that represent value throughout a delay. Despite this overall similarity, a few key differences were found. First, value was encoded less than 100 ms earlier in OFC compared to ACC. Second, value was more easily decoded in OFC immediately after the reward predicting cue, while a higher proportion of ACC neurons encoded value during the delay. These results argue in favor of the notion that value information is represented and processed earlier in the OFC than in ACC (*Kennerley and Wallis, 2009*). Nonetheless, the mixing of dynamic regimes appears quite similar between these two areas, so that both encode time sensitive and insensitive representations of value. Therefore, the underlying neural mechanisms responsible for encoding value over time are likely similar in OFC and ACC.

Overall, using analysis methods designed to extract different signal dynamics, we found that all co-exist to some extent within the same neuronal populations that represent the same task-related variable. Rich representations of expected values are critical for adaptive learning, and the mixed dynamics reported here could play an important role in conveying both general and temporally specific value expectations to areas involved in optimizing goal-directed behavior.

## Materials and methods

### Behavior and neurophysiology

The behavioral task has been previously described in *Rich and Wallis, 2016*; *Rich and Wallis, 2017*. Two head-fixed male Rhesus macaque monkeys (Macaca mulatta), aged 7 and 9 years, weighing 14 and 9 kg at the time of recording, were trained to perform a value-based decision making task in which they chose between visual stimuli associated with rewards. In the present study we focused on single cue trials only. All procedures were in accord with the National Institute of Health guidelines and recommendations of the University of California at Berkeley Animal Care and Use Committee. Subjects sat in a primate chair, viewed a computer screen and manipulated a bidirectional joystick. Task presentation and reward contingencies were controlled using MonkeyLogic software (*Asaad and Eskandar, 2008*), and eye movements were tracked with an infra-red camera (ISCAN, Woburn, MA).

A total of eight pictures (~2° × 3° of visual angle) comprised the set of stimuli associated with rewards. Pictures were selected randomly from this set on each trial. Four pictures predicted the delivery of juice reward (0.05, 0.10, 0.18, 0.30 ml), and four predicted that the length of a reward bar always present on the screen would increase by a set increment. Subjects were previously trained to associate the length of the bar to a proportional amount of juice obtained every four trials. Associations between cue and reward were probabilistic, so that on four out of seven trials the type and amount of reward was consistent with the cue. On one out of seven trials, the type of reward was different, on one out of seven trials, the amount was randomly picked to be different, and on one out of seven trials, both the amount and type differed from what the cue predicted.

Single cue trials were randomly interleaved with choice trials. Because the different reward values for primary (juice) and secondary (bar) outcomes were titrated, monkeys almost always chose the target associated with a higher value irrespective of the type of reward (*Rich and Wallis, 2016*).

During the delay between the reward predicting cue and reward delivery, monkeys were required to move a joystick either left or right depending on a visual cue, which we refer to as response instruction to avoid confusion with the reward predicting cue. Reward delivery was contingent on a correct joystick answer, and reaction times were inversely correlated with the size of the expected reward (*Rich and Wallis, 2016*).

Electrodes were lowered at the beginning of each session in the OFC (areas 11 and 13) and dorsal bank of the ACC sulcus (area 24). Recorded units were not screened for selectivity, but those with average activity lower than 1 Hz across the session were excluded. Further details on behavior and recording methods can be found in previous publications (*Rich and Wallis, 2016*; *Rich and Wallis, 2017*).

Firing rates of each unit were estimated with a 100ms-SD Gaussian kernel convolution over the spike train of entire recording sessions. The resulting firing rate was divided into epochs around the reward cue, the response (joystick) instruction cue and reward delivery, then binned into 100 ms bins

every 25 ms for encoding and every 50 ms for decoding methods (to reduce the number of computations, this includes cross-temporal decoding), so that there were 40 and 20 bins for 1 s of activity, respectively. Single and multi units from both monkeys were pooled together into a pseudo population for each region as the analyses presented in this paper did not produce different results between the two subjects (*Figure 2—figure supplement 1* and *Figure 5—figure supplement 3*). Note that significant decoding and encoding of value and type started before or on the presentation of reward predicting cue because of the smoothing method described above.

## Unit encoding

The firing rate of units was modeled with a linear regression with value, type of reward and their interaction as variables in the epochs described above. Value was defined as a categorical variable and analyzed with ANOVAs as the firing rate profile of single neurons did not always follow a linear trend with value (*Figure 2—figure supplement 2*). Individual F tests for each variable were applied on each time bin, and a bin of activity was considered to encode a variable if the p-value associated with the test was part of at least seven consecutive time bins (~175 ms with 25 ms steps between bins) where the p-value was lower than 0.01. Encoding strength was represented as the negative log of the F test p-value.

For both the sequential encoding and encoding duration analyses, trials were randomly split in half and the above encoding analysis was applied independently to each half during the delay activity, that is, from reward predicting cue offset to reward delivery. We only kept the neurons that had at least seven bins in a row with an encoding p-value less than 0.01 in both training and testing sets, hence the lower number of neurons in *Figure 3* and *Figure 4*. Peak encoding and encoding duration were obtained for each half of the data independently and compared across with Spearman correlations to test for their robustness.

## Decoding

Neurons across sessions and monkeys were pooled into a single population for each of OFC and ACC. Trials were randomly drawn from each session to match the number of trials in the population data. Because the choice of trials had an influence on the decoding accuracy, we repeated that draw five times to build five different data sets (Appendix 1, step [a]). The results of every population analysis presented in this paper is the average of the five randomly generated data sets. All reported accuracies of all decoding methods, including cross-temporal decoding, correspond to the average of a five-fold cross-validation on each of the five data sets, to ensure generalizability of the results (Appendix 1, step [b]).

Population representations of value were assessed with a ridge regression classifier (linear regression with $L_2$ 'ridge', or 'Tikhonov', regularization), which is a robust and efficient classifier as compared to SVM or logistic regression, which require costly computations. A one-versus-one approach was used to classify the four different classes of 'value', which is equivalent to six binary classifiers. The regularization parameter was optimized with a nested cross-validation within the outer five-fold cross-validation mentioned above by exploring every power of ten between −2 and 5 with a 0.5 step with the training data for each population or ensemble to maximize the average performance of the classifier trained and tested on each time point between cue onset and reward (Appendix 1, step [c]).

The significance of decoding accuracy was determined with permutation testing. The value label of trials was shuffled 1000 times and compared to the original data. The p-value was calculated as the number of permutations with a higher accuracy than the original data divided by 1000. p-Values across the five data sets were aggregated with an averaging method that does not require the p-values to be independent. The aggregated p-value was derived as two times the average of the five p-values from the five data sets (*Vovk and Wang, 2012*).

For cross-temporal decoding, we trained and tested the decoder on all possible combinations of time bin pairs. Accuracy along the diagonal represents training and testing at the same time point, while points away from the diagonal correspond to more temporally distant activities. Importantly, the testing data always correspond to out-of-sample trials so that the decoder was not trained and tested on the same trials, regardless of whether training and testing data points were from the same or different time bins.

## Subspace

To extract a stable value representation, we implemented the following method inspired by *Murray et al., 2017*. The firing rate activity of each neuron in a population was averaged over time from cue onset to reward delivery, and then averaged across the trials where the cue predicted the same value. The resulting matrix had four rows corresponding to the four values and n columns corresponding to the number of neurons in the population. A principal component analysis (PCA) was performed on this matrix and allowed us to project the neural population activity of each trial at every time bin onto a three dimensional subspace (four values minus one dimension). This method finds the dimensions that capture the highest variance related to value while discarding temporal information by averaging across time.

When decoding with the subspace, we used a nested cross-validation approach to ensure that the subspace was derived with data independent from the training and testing of the decoder to avoid inflated accuracy due to trial-wise noise (Appendix 1, step [e]). In other words, the training data was split into two sets of trials, one to define the subspace, the other to train the decoder which was then tested on the remaining trials. The assignment of the split data to either subspace or decoder training was then swapped and the decoder tested on the remaining data. As a result, this procedure elicited two matrices of accuracy for each fold of the inner cross-validation (Appendix 1, step [d]) and were averaged.

Permutation testing with the subspace shuffled only the labels of the training trials to more closely reproduce the propensity of the subspace trained with original data to elicit higher than chance accuracy.

## Ensembles

Testing all the possible combinations of units to find the ensemble that best decodes value is computationally infeasible. To remedy this *Backen et al., 2018* proposed a method that finds an ensemble that substantially optimizes decoding accuracy with a subset of the full population in a computationally feasible way, although it does not guarantee the selected ensemble is the single best. This method begins by screening each unit individually and selecting the most discriminating, then successively screens the remainder and adds the unit that most increases accuracy. Effectively, this method sorts the units in the order of highest contribution to decoding accuracy. However, decoding relies on the interaction of unit activities and by successively adding units, that method does not take into account all possible interactions. For example in a three neuron population, neuron 2 and 3 together might yield a higher decoding accuracy than any other pair of neurons, but neuron 1 alone yields the best accuracy and might be selected first, dismissing the possibility that neuron 2 and 3 are selected as the best 2-neuron ensemble. To partially circumvent this, we used the opposite method: successively removing units. As with the adding of units, we iteratively removed the units that maximized the decoding. This method yielded better results than adding the units.

In combination with the subspace method, stable ensembles were optimized with this method to find the ensembles that maximized the averaged cross-temporal decoding accuracy across all data points from cue onset to the reward delivery. This corresponds to averaging across all the points of the accuracy matrices represented in the figures except the points before cue and after reward. Once the best ensembles for each case (e.g. with or without subspace) were determined, cross-temporal decoding was performed with data beyond this period to show the boundaries of decoding accuracy. The dynamic ensemble was searched with the same iterative neuron-selection method that optimized a measure of temporal 'locality' of value representation as described in the main text.

For both stable and dynamic ensembles, the iterative ensemble optimization procedure involved a nested cross-validation procedure to ensure the independence of the testing and ensemble optimization trials (Appendix 1, step [d]). As a consequence, five ensemble searches were performed for each tested population and their corresponding testing accuracies were averaged.

To speed the computations associated with the stable ensemble search, a smaller number of bins was used by estimating the firing rate with non-overlapping 200 ms bins instead of the 100 ms bins with 25 ms steps used in the results. To achieve a similar result with the dynamic ensemble while keeping temporal precision of the data, one in five bins was included in the training while testing was performed on every bin.

## Unit and population measures

Three measures were defined to quantify the encoding of value in units. The encoding strength is the average of the negative log of the p-value across time during the delay:

$$es = \frac{1}{T}\sum_{t=1}^{T} -log(p(t)) \tag{1}$$

where $T$ is the total number of bins in the delay, and $p(t)$ is the p-value of a unit at time bin $t$ obtained from the ANOVA F-test described in the unit encoding section above.

The encoding duration is the number of bins where the above p-value was lower than 0.01 divided by the total number of bins.

The stability measure was defined as follow:

$$S = avg_{v,w}\left(\frac{1}{T}\left|\sum_{t=1}^{T} tanh\left(10 \times \frac{\overline{x_v(t)} - \overline{x_w(t)}}{sp_{vw}}\right)\right|\right) \tag{2}$$

where $\overline{x_v(t)}$ is the average firing rate of a unit for value $v$, $avg_{v,w}()$ is the average over all the possible combinations of values $v$ and $w$ except $v = w$, $tanh$ is the $]-1,1[$ bounded hyperbolic tangent function, $|\cdot|$ is the absolute value function and $sp_{vw}$ is the pooled standard deviation of the firing rate of a unit across trials with value $v$ and $w$:

$$sp_{vw} = \sqrt{\frac{s_v^2 + s_w^2}{2}} \tag{3}$$

where $s_v^2$ is the variance of the firing rate for trials with value $v$. Note that the difference of firing rate averaged by value, $\overline{x_v(t)} - \overline{x_w(t)}$, is signed and the measure takes the average of this signed difference. If a unit reverses its relative encoding of two values, the sign of the average difference changes over time and averaging across time will be close to zero, reflecting the lack of stability of encoding of that unit.

Two measures were used to quantify the contribution of units to the value subspace and the ensemble. For the value subspace, we used the absolute value of the subspace weights, which corresponds to the eigenvector components obtained through a PCA of the time and trial averaged data. The contribution of a unit to the stable subspace through the ensemble method was estimated by calculating the difference in average CTD accuracy (across all time bin pairs) by removing that unit from the ensemble. The last unit removed from the ensemble was discarded. For visualization of this specific measure, a symmetric square root transformation was applied where the absolute value of negative values was used before reassigning them their sign:

$$ac = sign(x)\sqrt{|x|} \tag{4}$$

The distribution of this measure across units was asymmetrical around the zero, so correlation analyses were applied individually on positive and negative contributions (see Main text).

A 'locality' measure was defined to optimize the ensemble of neurons eliciting a local CTD accuracy of value during the delay. We fit a Gaussian curve to the accuracies obtained from training at one time bin and testing on all delay bins:

$$f(x|\sigma,\mu,a,b) = a\exp\left(-\frac{(x-\mu)^2}{2\sigma^2}\right) + b \tag{5}$$

The offset value $b$ was fixed to 0.25, the chance level, and the mean of the Gaussian $\mu$ was set to the time of the training. Only the scaling factor $a$ and the standard deviation $\sigma$ were optimized. The locality measure $lm$ for a given ensemble is as follow:

$$lm = \frac{1}{T}\sum_{t_1=1}^{T} \frac{\max_{t_2}(\hat{f}_{t_1}) - .25}{\hat{\sigma}_{t_1}} \tag{6}$$

where $t_1$ and $t_2$ are the training time and testing time, respectively, $\max_{t_2}(\hat{f}_{t_1})$ is the maximum of the

Gaussian fitted on training at time $t_1$ and testing on all $t_2$ time bins, and $\hat{\sigma}_{t_1}$ is the standard deviation of the Gaussian fitted at time $t_1$. The contribution of a unit to the dynamic ensemble was calculated as the difference in locality measure when that unit is added to the ensemble.

## Correlations

To avoid inflated results from non-normally distributed data, we used Spearman correlations. In each of the figures showing these correlations, transformations (log or square) were applied for visualization purpose only. Note that these transformations do not affect the Spearman correlation, as it is based on rank and these transformations are monotonic and so preserve the rank of observations.

## Additional information

### Funding

| Funder | Grant reference number | Author |
|---|---|---|
| National Institute of Mental Health | R01-MH121448 | Joni D Wallis |
| National Institute of Mental Health | R01-MH097990 | Joni D Wallis |
| Hilda and Preston Davis Foundation | Postdoctoral fellowship | Erin L Rich |
| National Institute on Drug Abuse | K08-DA039051 | Erin L Rich |
| National Institute of Mental Health | R01-MH117763 | Joni D Wallis |
| Whitehall Foundation Research Grant | Research Grant | Erin L Rich |

The funders had no role in study design, data collection and interpretation, or the decision to submit the work for publication.

### Author contributions

Pierre Enel, Conceptualization, Software, Formal analysis, Validation, Investigation, Visualization, Methodology, Writing - original draft, Writing - review and editing; Joni D Wallis, Conceptualization, Resources, Supervision, Funding acquisition, Methodology, Project administration, Writing - review and editing; Erin L Rich, Conceptualization, Resources, Data curation, Supervision, Funding acquisition, Investigation, Methodology, Writing - original draft, Project administration, Writing - review and editing

### Author ORCIDs

Pierre Enel https://orcid.org/0000-0001-8983-6223
Erin L Rich https://orcid.org/0000-0002-7153-6027

### Ethics

Animal experimentation: This study was performed in strict accordance with the recommendations in the Guide for the Care and Use of Laboratory Animals of the National Institutes of Health (Assurance Number A3084-01). All of the animals were handled according to approved institutional animal care and use committee (IACUC) protocols (Protocol Number R283) of the University of California at Berkeley. All surgery was performed under isoflurane anesthesia, and every effort was made to minimize suffering.

### Decision letter and Author response

Decision letter https://doi.org/10.7554/eLife.54313.sa1
Author response https://doi.org/10.7554/eLife.54313.sa2

## Additional files

### Supplementary files

• Transparent reporting form

### Data availability

The neural recording data analyzed in this paper is available online at https://doi.org/10.5061/dryad.4j0zpc88b. The python scripts to reproduce the cross-temporal decoding analysis with subspace and ensembles is available online at https://github.com/p-enel/stable-and-dynamic-value (copy archived at https://github.com/elifesciences-publications/stable-and-dynamic-value).

The following dataset was generated:

| Author(s) | Year | Dataset title | Dataset URL | Database and Identifier |
|---|---|---|---|---|
| Enel P, Wallis JD, Rich EL | 2020 | Single- and multi-unit firing rate of two macaque monkeys' OFC and ACC neurons in value based decision-making task | https://doi.org/10.5061/dryad.4j0zpc88b | Dryad Digital Repository, 10.5061/dryad.4j0zpc88b |

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

# Appendix 1

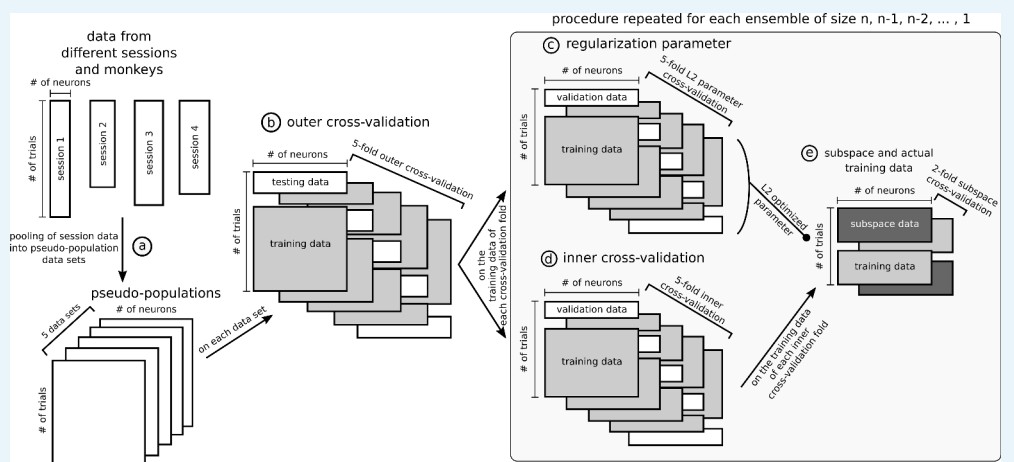

**Appendix 1—figure 1.** Data sets and cross-validation. In this figure, data matrices are represented with rectangles where rows are trials and columns units. Data from different sessions were pooled together into pseudo-populations by randomly selecting trials five times so that trial condition was matched (**a**) and formed five data sets. The rest of the figure represents the specific procedure used to optimize stable ensembles combined with the subspace method, which is the most involved nested cross-validation of the data and was applied independently on each of the five data sets. The outer 5-fold cross-validation (**b**) ensured that ensembles were not optimized and tested on the same data. The training data in each of the 5 folds of the outer cross-validation was used independently to optimize five stable ensembles (25 total, five data sets times five cross-validation folds). These data were further split following a 5-fold inner cross-validation to train and test a decoder on an ensemble (d) and used in parallel to optimize the $L_2$ normalization parameter of the ridge regression (**c**), with its own 5-fold cross-validation. To avoid defining the stable subspace and training the ridge decoder on the same data, the training trials from the inner cross-validation were further split in 2, one half for the subspace and the other for the data that were projected into the subspace and then used to train the decoder with the $L_2$ parameter optimized in step (**c**), and then this split was inversed to allow all data to be either part of the subspace or the training. Note that steps (**c**), (**d**) and (**e**) were repeated for each ensemble tested, from size n-1, n-2,... to a single unit. The ensemble eliciting the most stable representation for each fold of the outer cross-validation was tested on its corresponding test trials of the outer cross-validation. The cross-validation procedure for other analyses was similar but had fewer steps, for example the dynamic ensemble procedure did not involve the subspace/training data split.

