## [Decision Letter]

**Acceptance summary:**

In this study, Rich and colleagues recorded neural activity from the orbitofrontal and anterior cingulate cortex of monkeys during a delayed response task in which cues were presented that signaled different value primary or secondary reward. Their goal was to explore the representation of reward value during working memory performance to contrast predictions of proposals regarding whether information is maintained in a stable or dynamic manner. Using a novel approach of constructing ensembles only of neurons tailored to one or the other criteria, they argue persuasively that value information is available to downstream areas from the output of both regions at both timescales. The results are convincing and represent an important new perspective on this important question specifically and more generally on how to think about what information is represented in neural activity.

**Decision letter after peer review:**

Thank you for submitting your article "Stable and dynamic representations of value in the prefrontal cortex" for consideration by *eLife*. Your article has been reviewed by three peer reviewers, including Geoffrey Schoenbaum as the Reviewing Editor and Reviewer #3, and the evaluation has been overseen Kate Wassum as the Senior Editor. The following individual involved in review of your submission has agreed to reveal their identity: Laurence Tudor Hunt (Reviewer #2).

The reviewers have discussed the reviews with one another and the Reviewing Editor has drafted this decision to help you prepare a revised submission.

Summary:

In this study, Rich and colleagues recorded neural activity from the OFC and ACC of monkeys during a delayed response task in which cues were presented that signaled different value primary or secondary reward. Their goal was to explore the representation of reward value during working memory performance to contrast predictions of proposals regarding whether information is maintained in a stable or dynamic manner. Using a novel approach of constructing ensembles only of neurons tailored to one or the other criteria, they argue persuasively that value information is available to downstream areas from the output of both regions at both timescales. The results are convincing and represent an important new perspective on this important question specifically and more generally on how to think about what information is represented in neural activity.

Essential revisions:

The reviewers were each positive about the research and the paper, in the reviews and in the discussion. In discussion, there were four essential revisions noted below that are most important.

– The first is to clarify why decoding in Figure 5 is high prior to cue onset in some panels. This was commented on by two of the reviewers and must be addressed.

– The second is the issue of how neurons were selected and the issues of cross-validation and circularity raised by these same reviewers.

– Related to this is a third issue, which is whether or not neurons were included as value encoding and in the subsequent ensemble analyses if they had categorical or non-linear value correlates. The authors mention this was true, and on discussion it was felt that it was important to clarify this and ideally to show that the results either do or do not require their inclusion with more traditional linear correlates.

– And finally, fourth, all reviewers agreed that a parallel analysis considering type would be of interest. It is not critical what it shows, but it was felt this analysis could be done and might affect how some interpret these results.

Further details on these points can be found in the full reviews.

Reviewer #1:

This study tests how single and multi-unit recordings from ACC and OFC of non-human primates encode value information across delays. The authors find that both value and reward type information can be decoded from activity in both areas. In addition, subspace analyses for value information show that encoding is both stable and dynamic across time.

The nature of encoding across delays is a timely topic. Here the authors consider both the stability and the dynamics of this encoding over time. This is innovative and I think the results tell us a lot about the nature of neural coding in OFC/ACC. I have a few concerns that should be addressed.

1) I think it is a missed opportunity that the more interesting analyses are only applied to value but not reward type. Encoding of reward type has received considerable attention recently (e.g., Stalnaker et al., 2014; Howard et al., 2015), and here, decoding accuracy for value and type was nominally comparable, so it is unclear why this wasn't done here. I'd strongly encourage the authors to apply the analyses focusing on stability and dynamics of encoding to reward type as well.

2) I do not understand how it is possible that, as shown in Figure 5B (lower panel), the decoder trained on data from 500 ms before cue onset (that is before neurons encode any value) can decode value throughout the rest of the trial.

3) Related to the point above, it appears there are at least two potential sources of non-independence errors in the decoding analyses.

3.1) First, it appears the L2 parameter was selected based on all data and that "the parameter eliciting the best decoding for each condition was used for all decoding results presented here". As described this analysis is circular and will inflate decoding accuracies. Instead the authors should use nested cross-validation, where the parameter is optimized (using cross-validation) within the training data. (That is, the data is split into 3 parts. Different classifiers with different levels of L2 are trained on part 1. These classifiers are then evaluated on part 2 and accuracies are computed. The value of L2 that results in the best accuracy (determined in part 2) is then used to train a classifier based on part 1 and 2, and the final accuracy is computed based on part 3. No parameter decisions are ever based on accuracy that is determined using part 3)

3.2) The second instance of non-independence may occur during the ensembles selection approach. This is not fully explained, but it appears that the ensemble selection was not based on nested cross-validation either. Again, a nested cross validation should be used such that the ensembles are selected within the training data using cross-validation, and the final accuracy matrices should be computed based on data that was never used in the selection procedure.

Reviewer #2:

This study provides further insight into the recent debate around 'stable' versus 'dynamic' representations of value in prefrontal cortex. In short, the authors find evidence supporting both types of value representation. The authors use a range of methods to look at this, but perhaps the most convincing analyses are shown in Figure 5, where different subsamples of the same neural data (different 'ensembles') can be used to create cross-temporal decoding patterns that either appear 'stable' or 'dynamic', with the original data containing a mixture of the two.

Overall I found the paper to provide good support for a hybrid of stable and dynamic coding schemes, and the data to be well analysed. However, there were several points in the paper that I thought could do with a bit more attention.

1) Figure 2A vs. 2C Why are the % of encoding neurons broadly similar between ACC and OFC, but the decoding accuracy different between the two regions shortly after cue onset? This is a bit of an outstanding question that the authors don't really address.

2) In Figure 3, the authors do the right thing of splitting their data into training/test data when sorting by peak latency. (Otherwise, the tiling can easily arise from random noise, e.g. https://twitter.com/steinmetzneuro/status/1150971584812920832). It is, however, equally important that they do this in Figure 4.

3) "We considered value as a categorical variable, as some units activated in a non-linear fashion, for example by responding only to a specific value and could not be modeled with a simple linear regression" – isn't it possible that these neurons are encoding picture identity rather than value? In order to show that there is non-linear coding of value, then presumably the prediction would be that this non-linear value coding would generalise across the primary and secondary reinforcers? This could be easily shown for Supplementary Figure 2 – the lines could be split into primary and secondary reinforcer, with the 'non-linear value' prediction meaning that these two lines should overlap.

4) Figure 5 – it looks like the ACC cross-temporal decoding in the stable subspace goes back in time to before the stimulus onset (i.e. the classifier trained from timepoints -500ms to 0ms pre-cue has above-chance decoding accuracy). Is there a reason for this?

5) I couldn't understand how the 'trajectory speed' in Figure 6A is calculated – is the 'distance' in neural space being computed on raw firing rates, or on classification accuracies? Some further detail on this would be helpful.

6) It would also be interesting to know whether the regression shown in Figure 6B is less successful in the stable ensemble than in the dynamic ensemble?

7) "While value representations have been found in the activity of ACC and OFC e.g. [Rushworth and Behrens, 2008], this is, to our knowledge, the first study of their dynamics during a short delay." – It might be relevant to cite Cavanagh et al., 2016 here? That study used cross-temporal correlation to study how value coding persisted during the delay between the choice and outcome periods, albeit in the context of different subpopulations with different resting autocorrelations (see their Figure 5).

Reviewer #3:

In this study, Rich and colleagues recorded neural activity from the OFC and ACC of monkeys during a delayed response task in which cues were presented that signaled different value primary or secondary reward. Their goal was to explore the representation of reward value during working memory performance – across a delay period in the current task – to contrast predictions of proposals regarding how information regarding expected value would be maintained – whether it would be stable or dynamic. Using a novel approach of hand-picking neurons to compose ensembles tailored to one or the other criteria, they argue persuasively that value information is available to downstream areas at both timescales in both of these two prefrontal areas. The results are convincing I think and represent an important new perspective on this important question specifically and more generally on how to think about what information is represented in neural activity. So I am very positive generally about this paper. That said, I do have some things I would really like to see done.

1) I did not see any behavior. I think it is important to show the behavior in this paper in order to demonstrate that the monkeys do treat the cues as if they have these scalar values and further to show that they treat the cues predicting putatively similar amounts of primary and secondary reward similarly. The latter may be particularly important wrt to my second point below. This should be shown using choice behavior on the probe trials, but I also think it should be evident in reaction time or some other measure during the forced choice trials that comprise the neural data set.

2) I found the classification that something is encoding value to be confusing. The authors say that value was treated as a categorical variable since some neurons had idiosyncratic responses to cues not at the ends of the value ranges. Yet it seems to me that this begs the question of whether these neurons are signaling value. At a minimum, to count as a value neuron, a cell should show the same response across both trial types that have the same putative value. So a value neuron must either have a linear value correlate or it would need to fall into this special category… and even then, I think calling such a cell a "value cell" stretches the conventional definition of value. For instance, if a neuron fired the most to 2 and then significantly less to 1, 3, and 4, I would have trouble seeing this as a value cell by most definitions of value, even if it fired the same to 2 on both trial types. Given this, in my opinion, analyses such as those in Figures 2 and 5 that are the heart of the paper need to be redone without such "non-linear" value neurons as part of the populations. If the results only hold with those neurons included, then I think the interpretation of what is being represented statically and dynamically might be significantly affected for me.

3) Looking at Figure 2C and D, I do not understand the statement at the end of the subsection “Encoding and decoding of value in unit activity”, that because reward type is poorly encoded, the paper focuses on value. I think it is ok to focus on value because that is the question of interest. But this statement misrepresents the reason it seems to me. And really I'd like to see both explored either separately or as part of the same analysis. Both information is relevant to the broader question I think; primary versus secondary is as interesting a part of what is being maintained as simple value. In fact I'd say it is arguably more important, since it is important to me to be paid in dollars and not equivalent piles of marshmallows for example!?!

---

## [Author Response]

Essential revisions:The reviewers were each positive about the research and the paper, in the reviews and in the discussion. In discussion, there were four essential revisions noted below that are most important.– The first is to clarify why decoding in Figure 5 is high prior to cue onset in some panels. This was commented on by two of the reviewers and must be addressed.

We thank the reviewers for identifying this problem. After some investigation, we uncovered several contributors to this result, which we describe below, all of which have been addressed in the revised manuscript. Given that this issue turned out to arise from some unexpected consequences of the analytic approaches, we include details of our revised methods in the manuscript along with a supplementary figure to inform the field.

The first problem we identified was that the decoder was trained with the same data that was used to define the value subspace, making it possible for trial-wise noise to bias the pre-cue activity once it was projected onto the subspace. To remedy this issue, we divided the training data in two halves; the first was used to define the subspace and the other to train the decoder. We then repeated the process using the first half for the decoder and the second half for the subspace so that all data was used in turn for subspace and training. We averaged across the two accuracy matrices to obtain the final result. Updated Materials and methods section:

“When decoding with the subspace, we used a nested cross-validation approach to ensure that the subspace was derived with data independent from the training and testing of the decoder to avoid inflated accuracy due to trial-wise noise (Appendix 1—figure 1E). […] As a result, this procedure elicited two matrices of accuracy for each fold of the inner cross-validation (Appendix 1—figure 1D) and were averaged.”

Because of this sensitivity to bias, we also cross-validated the selection of trials that were part of the analysis. That is, the neural population studied here is composed of several sessions and subjects pooled together, and trials were randomly selected from these to arrive at the same number of trials per condition. To create a non-biased decoding estimate, we averaged the accuracy matrices of 5 random selections of trials. Updated Materials and methods section:

“Neurons across sessions and monkeys were pooled into a single population for each of OFC and ACC. […] The results of every population analysis presented in this paper is the average of the 5 randomly generated data sets.”

Finally, we adapted permutation testing to the subspace method. We noticed that small fluctuations in the activity before the cue had the potential to change decoding accuracy when the decoder was trained on these data and tested after the cue. This indicated that the decoder was simply parsing normal variance in the neural activity, which should be addressed by appropriate permutation tests. To address this in the cross-temporal decoding with the subspace method, instead of shuffling all the trials, the subspace and testing trials were kept as is and only the training trials were shuffled. This method successfully reproduced the divergence from chance accuracy with the original data, and thus prevents spurious decoding before the cue to reach significance. Corresponding updated Materials and methods:

“Permutation testing with the subspace shuffled only the labels of the training trials to more closely reproduce the propensity of the subspace trained with original data to elicit higher than chance accuracy.”

The reviewers will still notice significant decoding a few ms before the cue in the cross-temporal decoding, but this is now merely due to smoothing windows, and is absent from bins that contain only pre-cue data. We kept the original longer smoothing windows, as these are more efficient for decoding in our experience, and elicit smoother accuracies making interpretations easier for the timescale of our investigations. This point is mentioned in the legend of the cross-temporal decoding figure:

“[…]. Note that significant decoding before the presentation of the cue is due to smoothing.”

The splitting of the data is illustrated in a new figure (Appendix 1—figure 1):

“Data sets and cross-validation. In this figure, data matrices are represented with rectangles where rows are trials and columns units. […] The cross-validation procedure for other analyses was similar but had fewer steps, e.g. dynamic ensemble procedure did not involve the subspace/training data split.”

– The second is the issue of how neurons were selected and the issues of cross-validation and circularity raised by these same reviewers.

Thank you for bringing this issue to our attention. We now select neurons to be part of an ensemble with a cross-validation method, so the final accuracy now reflects the average of a 5-fold cross-validation that resulted in 5 different ensembles and their corresponding accuracies. The final cross-temporal decoding method now includes several levels of nested cross-validation as described in the updated Materials and methods section:

“For both stable and dynamic ensembles, the iterative ensemble optimization procedure involved a nested cross-validation procedure to ensure the independence of the testing and ensemble optimization trials (Appendix 1—figure 1D). As a consequence, 5 ensemble searches were performed for each tested population and their corresponding testing accuracy were averaged.”

– Related to this is a third issue, which is whether or not neurons were included as value encoding and in the subsequent ensemble analyses if they had categorical or non-linear value correlates. The authors mention this was true, and on discussion it was felt that it was important to clarify this and ideally to show that the results either do or do not require their inclusion with more traditional linear correlates.

Indeed, our encoding and decoding methods consider value as a categorical variable and we recognize that this choice was not discussed in detail in the manuscript. We also agree that it is a departure from the traditional perspective that considers value coding to be a monotonic relationship between activities in single units and reward quantities, and now discuss this in a bit more detail by including the following text:

“We considered value as a categorical variable, as some units activated in a non-linear fashion with respect to value, for example by responding only to a specific value, and could not be modeled with a simple linear regression (Figure 2—figure supplement 2). […] Approximately 13% of units in both OFC and ACC fit these criteria.”

(Also relevant to this discussion is the response to reviewer 2’s comment below on distinguishing non-linear responses to value, versus responses to an individual picture.)

The larger issue raised by this comment, however, is whether the results that we report depend on the inclusion of non-linear coding neurons. To address this, we evaluated all neurons with a simple linear regression that included value, type and their interaction as independent variables and re-processed all the data with only linear neurons, which we defined as those that were significant for the value regressor for at least 7 bins in a row (~175ms) at any time from the onset of cue presentation to the onset of reward delivery. Note that a neuron that is classified as linear because of a significant value coefficient in a linear regression model at one time point, might not be significant at another time point with this test but will be with an ANOVA instead, so it is particularly difficult to cleanly parse populations of linear vs. non-linear neurons. However, this approach was designed to capture neurons with traditional monotonic value coding with methods typically used in the literature. Overall, all results were qualitatively similar when analyses were performed with only linear neurons defined in this fashion (see Author response images 1-3). This confirms that our results are not dependent on the presence of non-linear value coding neurons, and leaves unchanged our main interpretations.

**Author response image 1. sa2fig1:** Figure 2, linear value neurons only.

**Author response image 2. sa2fig2:** Figures 3 and 4, linear value neurons only.

**Author response image 3. sa2fig3:** Figure 5, linear value neurons only.

– And finally, fourth, all reviewers agreed that a parallel analysis considering type would be of interest. It is not critical what it shows, but it was felt this analysis could be done and might affect how some interpret these results.

All the main analyses (sequential and persistent encoding, and cross-temporal decoding) are now also done on “type” variable and are included in the paper: Figure 2 was revised (panel F was added), and Figures 3—figure supplement 1, Figure 4—figure supplement 1 and Figure 5—figure supplement 1 was added.

Further details on these points can be found in the full reviews.Reviewer #1:[…] The nature of encoding across delays is a timely topic. Here the authors consider both the stability and the dynamics of this encoding over time. This is innovative and I think the results tell us a lot about the nature of neural coding in OFC/ACC. I have a few concerns that should be addressed.1) I think it is a missed opportunity that the more interesting analyses are only applied to value but not reward type. Encoding of reward type has received considerable attention recently (e.g., Stalnaker et al., 2014; Howard et al., 2015), and here, decoding accuracy for value and type was nominally comparable, so it is unclear why this wasn't done here. I'd strongly encourage the authors to apply the analyses focusing on stability and dynamics of encoding to reward type as well.

As noted in the combined comments section, we have now included analyses for reward type. In addition, we now include the following notes discussing the encoding of reward type:

“While value signals were robust in both OFC and ACC, consistent with previous reports of strong abstract value coding in these regions (Padoa-Schioppa, 2006, Rushworth, 2008), there were also weaker representations of the anticipated reward type. This is consistent with findings in human and rodent OFC reporting value signals that are specific to a particular type of outcome (Howard, 2015, Stalnaker, 2014).”

2) I do not understand how it is possible that, as shown in Figure 5B (lower panel), the decoder trained on data from 500 ms before cue onset (that is before neurons encode any value) can decode value throughout the rest of the trial.

Answered above in the combined comments section.

3) Related to the point above, it appears there are at least two potential sources of non-independence errors in the decoding analyses.3.1) First, it appears the L2 parameter was selected based on all data and that "the parameter eliciting the best decoding for each condition was used for all decoding results presented here". As described this analysis is circular and will inflate decoding accuracies. Instead the authors should use nested cross-validation, where the parameter is optimized (using cross-validation) within the training data. (That is, the data is split into 3 parts. Different classifiers with different levels of L2 are trained on part 1. These classifiers are then evaluated on part 2 and accuracies are computed. The value of L2 that results in the best accuracy (determined in part 2) is then used to train a classifier based on part 1 and 2, and the final accuracy is computed based on part 3. No parameter decisions are ever based on accuracy that is determined using part 3)

Indeed, the regularization parameter was not optimized in a fully independent way and we have now corrected this. The L2 parameter is now optimized on the training data only with a nested cross-validation. For ensemble optimization, this parameter was not optimized independently for each possible ensemble n that was explored but for the ensemble n+1 that was previously selected in the iterative search to limit the computations as this parameter search is computationally expensive, but because the n and n+1 datasets were almost identical, the potential loss in accuracy due to a suboptimal L2 parameter selection should be minimal. The Materials and methods section was updated as well to reflect this change:

“The regularization parameter was optimized with a nested cross-validation within the outer 5-fold cross-validation mentioned above by exploring every power of ten between -2 and 5 with a 0.5 step with the training data for each population or ensemble to maximize the average performance of the classifier trained and tested on each time point between cue onset and reward (Appendix 1—figure 1C).”

3.2) The second instance of non-independence may occur during the ensembles selection approach. This is not fully explained, but it appears that the ensemble selection was not based on nested cross-validation either. Again, a nested cross validation should be used such that the ensembles are selected within the training data using cross-validation, and the final accuracy matrices should be computed based on data that was never used in the selection procedure.

Answered above in the combined comments section.

Reviewer #2:[…] Overall I found the paper to provide good support for a hybrid of stable and dynamic coding schemes, and the data to be well analysed. However, there were several points in the paper that I thought could do with a bit more attention.1) Figure 2A vs. C Why are the % of encoding neurons broadly similar between ACC and OFC, but the decoding accuracy different between the two regions shortly after cue onset? This is a bit of an outstanding question that the authors don't really address.

This is indeed an interesting point that was not fully explored in the paper. Now that decoding is performed on 5 different data sets and the resulting accuracies averaged, the difference in decoding accuracy between ACC and OFC is smaller yet still observable. This difference is partly explained by a higher number of units in the OFC population, because subsets of OFC units matched to the number of ACC units elicit lower accuracy, as shown in Figure 2C and D. However, there is still a gap in accuracy even after accounting for the difference in population size, and this is likely due to the fact that the two analyses rest on different measures – either the proportion of neurons that meet criteria for significantly encoding value, or the quality of this value signal in the population. The difference between these measures suggests that there is a higher fidelity value signal in OFC at the population level, despite slightly more neurons in ACC meeting our encoding criteria.

To support this perspective, we looked closer at the encoding p-values in each region. While the distributions are indistinguishable (p-values of encoding value from ANOVA at peak encoding [325ms after cue], distributions compared with Kruskall-Wallis test [K=.62, pval=.43]), sensitivity indices (also known as d’) are higher on average in OFC than ACC (average across neurons and values at peak encoding: OFC=0.26, ACC=0.22). This supports the view that the same proportion of neurons encode value after the cue presentation, but the way value is encoded has an influence on the decoder. Thus, in this case, a single value obtained from a linear model, whether it is a linear regression or an ANOVA, has limited ability to describe the encoding of complex information like value.

A shortened version of this discussion has been added to the Results:

“Both regions had similar proportions of value encoding after the cue presentation, however decoding accuracy was more than 10% higher in OFC, which can be partly explained by a higher unit count in the OFC population. […] The sensitivity index averaged across pairs of values in all neurons was higher in OFC (0.26) than in ACC (0.22), suggesting that values might be easier to separate in OFC activity compared to ACC, leading to higher decoding accuracy.”

2) In Figure 3, the authors do the right thing of splitting their data into training/test data when sorting by peak latency. (Otherwise, the tiling can easily arise from random noise, e.g. https://twitter.com/steinmetzneuro/status/1150971584812920832). It is, however, equally important that they do this in Figure 4.

Thank you for pointing out this issue, we applied the cross-validation method to the encoding duration of single neuron and found that encoding duration is a robust feature as well. It is now reflected in the encoding durations in Figure 4 and the main text in the Results section:

*“*However, sorting the neurons by the proportion of the delay during which they encode value offers a different picture. Figure 4 shows the same splitting procedure applied to encoding duration (Figure 4—figure supplement 1 for variable type).”

And in the Materials and methods section:

“For both the sequential encoding and encoding duration analyses, trials were randomly split in half and the above encoding analysis was applied independently on each half on the delay activity, i.e. from reward predicting cue offset to reward delivery.”

3) "We considered value as a categorical variable, as some units activated in a non-linear fashion, for example by responding only to a specific value and could not be modeled with a simple linear regression" – isn't it possible that these neurons are encoding picture identity rather than value? In order to show that there is non-linear coding of value, then presumably the prediction would be that this non-linear value coding would generalise across the primary and secondary reinforcers? This could be easily shown for Supplementary Figure 2 – the lines could be split into primary and secondary reinforcer, with the 'non-linear value' prediction meaning that these two lines should overlap.

As noted by the reviewer, this task is well constructed to determine whether non-linear neurons simply respond to one picture from the set, potentially encoding picture identity, or whether they respond to (e.g.) two pictures corresponding to one of the four possible values and therefore are code value in a non-linear fashion, regardless of picture or reward type. To address this, we followed the reviewer’s suggestion and Figure 2—figure supplement 2 now shows examples of non-linear neurons with activities separated by reward type showing non-linear value encoding that is similar for primary and secondary rewards. To explore this question quantitatively, we focused on activity at the peak of encoding (325 ms after cue presentation) and defined a unit as non-linear value coding if it met these 3 criteria:

– the unit *was not* significant for value with a linear regression described above in the response to combined comments (activity ~ value + type + value x type)

– the unit *was* significant for value with an ANOVA (activity ~ C(value) + type + value x type)

– the unit *was not* significant for the interaction of value and type with an ANOVA (activity ~ C(value) + type + value x type)

With this method we found that approximately *13% of neurons* in both OFC and ACC were non-linearly encoding value. Unfortunately, since there is only a single cue per value/type combination, it is impossible to further parse neurons that show an interaction in the above analyses into responses that are truly selective for picture identity versus those that encode the value of a specific outcome (i.e. the interaction of value and type), however we found with visual inspection of firing rate profiles that potential picture neurons encoding a single combination of value and type were very rare. These results have been added to the manuscript in the text identified in the combined comments section above.

4) Figure 5 – it looks like the ACC cross-temporal decoding in the stable subspace goes back in time to before the stimulus onset (i.e. the classifier trained from timepoints -500ms to 0ms pre-cue has above-chance decoding accuracy). Is there a reason for this?

Answered above in the combined comments section.

5) I couldn't understand how the 'trajectory speed' in Figure 6A is calculated – is the 'distance' in neural space being computed on raw firing rates, or on classification accuracies? Some further detail on this would be helpful.

The trajectory speed was calculated from the firing rate of individual units estimated with the same method used for the rest of the paper, albeit with a smaller smoothing window for more temporally accurate data. This was previously only explained in the legend of Figure 6, so to improve clarity we have now also indicated this in the main text as follows:

“This dynamic pattern was also reflected in the population trajectory speed obtained by calculating the distance between two successive time bins in the neural space spanned by the estimated firing rate of each unit with a shorter smoothing window for more temporally accurate firing rate (50 ms SD Gaussian kernel […]”6) It would also be interesting to know whether the regression shown in Figure 6B is less successful in the stable ensemble than in the dynamic ensemble?

Figure 6B showed a regression of time bins against the predicted time bin based on neural activity in the dynamic ensemble. The reviewer’s suggestion is that this should better predict time than the stable ensemble, which we found was generally the case (Author response image 4). The sum of squared errors was higher for the stable ensembles than for the dynamic ensemble, however the original population still predicted time better than either of these two ensembles (SSE: original population = 9018, dynamic ensemble = 26851, stable ensemble = 36309), suggesting the presence of additional temporal information that is not present in either reduced ensemble.

**Author response image 4. sa2fig4:** 

7) "While value representations have been found in the activity of ACC and OFC e.g. [Rushworth and Behrens, 2008], this is, to our knowledge, the first study of their dynamics during a short delay." – It might be relevant to cite Cavanagh et al., 2016 here? That study used cross-temporal correlation to study how value coding persisted during the delay between the choice and outcome periods, albeit in the context of different subpopulations with different resting autocorrelations (see their Figure 5).

Thank you for this reference, this reference has been added to the Discussion in two sections:

“In this study, we shed light on the complex dynamics of value representations in prefrontal cortex. While value representations have been found in the activity of ACC and OFC e.g. [Rushworth, 2008a], little is known about their dynamics across delays. […] Here, we expand on this to show that targeted methods elicit seemingly opposite results…”

“These results present the coding of value in two extreme dynamical regimes, either fully stable or fully dynamic. […] Indeed, it has been shown that the time constant of neurons encoding value in prefrontal region is diverse [Cavanagh et al., 2016], supporting the hypothesis of multitude of dynamical regimes.”

Reviewer #3:[…] 1) I did not see any behavior. I think it is important to show the behavior in this paper in order to demonstrate that the monkeys do treat the cues as if they have these scalar values and further to show that they treat the cues predicting putatively similar amounts of primary and secondary reward similarly. The latter may be particularly important wrt to my second point below. This should be shown using choice behavior on the probe trials, but I also think it should be evident in reaction time or some other measure during the forced choice trials that comprise the neural data set.

In the previous manuscript, behavioral results were only briefly mentioned, and the reader was referred to a previous publication for further description. To enhance the current presentation, Figure 1 has now been extended (Figure 1C) to show the preference of the monkeys for higher valued cues with little influence of the type of reward. These results were quantified with a logistic regression model on choice trials, in which the monkeys chose between two cues. In addition, a linear regression model predicting the reaction time of monkeys in the joystick task shows a significant effect of value but not type. The following text has been added:

Figure 1C legend: “Probability of choosing a juice option in choice trials for every pair of cues in which one predicts a juice reward and the other a bar reward.”

“A logistic regression model predicting the cues chosen by the animals with variables value and type shows a much lower coefficient for type, and its p-values are many order of magnitude larger, confirming that the type of reward had little influence on the choice of animals compared to value (Figure 1C, coefficients with [p-value of t-test] for monkeys M and N respectively; type left/right:.41 [2.53e-3] / -.51 [1.75e-4] and -.61 [2.6e-3] /.32 [0.11], value left/right: 1.77 [6.1e-117] / -1.95 [3.02e-136] and 2.66 [1.59e-94] / -2.46 [2.27e-90]).”

“Reaction times were also unaffected by the type of reward anticipated (linear regression predicting reaction time as a function of value and type in single cue trials : coefficients for value [p-value of t-test] for monkeys M and N respectively; type: -21 [.08] and 14.5 [.32], value: -63.9 [1.73e-31] and -65.7 [2.66e-23]).”

2) I found the classification that something is encoding value to be confusing. The authors say that value was treated as a categorical variable since some neurons had idiosyncratic responses to cues not at the ends of the value ranges. Yet it seems to me that this begs the question of whether these neurons are signaling value. At a minimum, to count as a value neuron, a cell should show the same response across both trial types that have the same putative value. So a value neuron must either have a linear value correlate or it would need to fall into this special category… and even then, I think calling such a cell a "value cell" stretches the conventional definition of value. For instance, if a neuron fired the most to 2 and then significantly less to 1, 3, and 4, I would have trouble seeing this as a value cell by most definitions of value, even if it fired the same to 2 on both trial types. Given this, in my opinion, analyses such as those in Figure 2 and 5 that are the heart of the paper need to be redone without such "non-linear" value neurons as part of the populations. If the results only hold with those neurons included, then I think the interpretation of what is being represented statically and dynamically might be significantly affected for me.

Answered above in the combined comments section.

3) Looking at Figure 2C and D, I do not understand the statement at the end of the subsection “Encoding and decoding of value in unit activity”, that because reward type is poorly encoded, the paper focuses on value. I think it is ok to focus on value because that is the question of interest. But this statement misrepresents the reason it seems to me. And really I'd like to see both explored either separately or as part of the same analysis. Both information is relevant to the broader question I think; primary versus secondary is as interesting a part of what is being maintained as simple value. In fact I'd say it is arguably more important, since it is important to me to be paid in dollars and not equivalent piles of marshmallows for example!?!

Answered above in the combined comments section.